# Countering Backdoor Attacks in Image Recognition: A Survey and Evaluation of Mitigation Strategies

**Kealan Dunnett**                                              *kealan.dunnett@hdr.qut.edu.au*
*Queensland University of Technology*
*Brisbane, Australia*

**Reza Arablouei**
*Commonwealth Scientific and Industrial Research Organisation*
*Brisbane, Australia*

**Dimity Miller**
*Queensland University of Technology*
*Brisbane, Australia*

**Volkan Dedeoglu**
*Commonwealth Scientific and Industrial Research Organisation*
*Brisbane, Australia*

**Raja Jurdak**
*Queensland University of Technology*
*Brisbane, Australia*

**Reviewed on OpenReview:** *https://openreview.net/forum?id=OysA7cuCUh*

## Abstract

The widespread adoption of deep learning across various industries has introduced substantial challenges, particularly in terms of model explainability and security. The inherent complexity of deep learning models, while contributing to their effectiveness, also renders them susceptible to adversarial attacks. Among these, backdoor attacks are especially concerning, as they involve surreptitiously embedding specific triggers within training data, causing the model to exhibit aberrant behavior when presented with input containing the triggers. Such attacks often exploit vulnerabilities in outsourced processes, compromising model integrity without affecting performance on clean (trigger-free) input data. This paper surveys and benchmarks backdoor-mitigation methods for image classification, where the defender has access to a compromised model and a small clean mitigation set but no trigger or poisoned data. We review 23 methods and benchmark 19 of them against eight attacks across three datasets, four architectures, three poisoning ratios, and three mitigation-data regimes, yielding over 120,000 individual experiments. The results show that many methods reduce the attack success rate (ASR) in some settings, but improvements over the Fine-Pruning (FP) and fine-tuning (FT) baselines are modest and inconsistent. In particular, low ASR often comes without recovery of the correct label for triggered samples. We synthesize recurring failure modes (data-limited overfitting, assumption-driven inconsistency, recovery accuracy collapse, and hyperparameter brittleness) and outline challenges for developing mitigation methods that are practical and robust across attacks, architectures, and datasets.

## 1 Introduction

Deep learning models now support decision-making in domains such as healthcare, education, transportation, and logistics Pouyanfar et al. (2018). Their ability to extract complex patterns from data has driven this

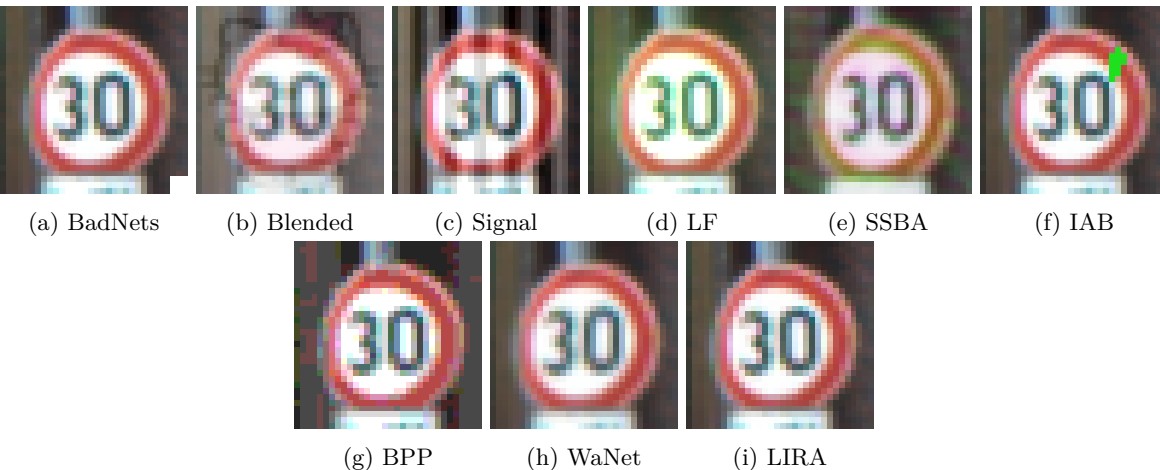

(a) BadNets    (b) Blended    (c) Signal    (d) LF    (e) SSBA    (f) IAB

(g) BPP    (h) WaNet    (i) LIRA

Figure 1: Examples of different backdoor triggers used in the literature. Note that while IAB adds a local patch to each image, its position and scale can vary across images.

adoption, but it also makes their behavior difficult to interpret and secure. This opacity creates opportunities for adversaries to manipulate model behavior in ways that are hard to diagnose before deployment Wang et al. (2019c).

Image-classification models are known to be vulnerable to adversarial manipulation. The seminal work of Szegedy et al. (2013) showed that small, carefully constructed input perturbations can cause large prediction errors, and subsequent work has identified a wide range of related threats across learning tasks Wang et al. (2019c). Backdoor attacks are a particularly important training-time threat because they compromise a model before deployment while leaving its clean-input performance largely intact Gu et al. (2017).

A backdoor attack intentionally creates an association between a spurious input feature, known as a trigger, and an attacker-specified classification outcome. Once this association is learned, a compromised model behaves normally on clean images but produces the backdoor behavior when the trigger is present. In this way, backdoor attacks undermine the integrity of model decision-making by inserting an unwanted task without substantially degrading the model's ability to perform the original clean-image classification task Li et al. (2022).

A successful backdoor attack requires the adversary to compromise some part of the victim's training pipeline. This threat is realistic when training data are collected by third parties, when models are trained through cloud-based Machine-Learning-as-a-Service (MLaaS) platforms Liu et al. (2019), or when practitioners initialize training from third-party pretrained weights. In all of these settings, the adversary may be able to influence the data, the training procedure, or the initial model without the defender's knowledge. The widespread use of pretrained models makes this concern practical rather than purely theoretical: a recent survey reported that 48.1% of participants use third-party weights for model training Grosse et al. (2023). Popular model-sharing and MLaaS platforms further amplify this attack surface.

Machine learning security aims to develop methods that protect models against such threats Yuan et al. (2019). Within this field, backdoor mitigation seeks to remove backdoor behavior from a compromised model while preserving clean-task performance. Despite substantial recent activity, the reliability of current mitigation methods remains uncertain. Many methods are evaluated over a limited set of attacks, datasets, architectures, or mitigation-data regimes, making it difficult to judge their practical robustness. Moreover, many evaluations emphasize only attack success rate (ASR). ASR reduction is necessary, but it is not sufficient: if triggered inputs are no longer classified as the attacker's target but are still not restored to their correct class, the defense has not recovered the original task behavior. Recovery accuracy (RA), and its normalized form RDR, therefore need to be considered alongside ASR.

Table 1: Comparison of our work with existing related surveys. *# Eval.* = number of mitigation proposals benchmarked. *Data Avail.* = whether mitigation-data availability is varied across settings.

| Reference | Year | Domain Tasks | Defensive Eval. | Empirical Eval. | # Methods | Data Avail. | Multiple Architectures |
|---|---|---|---|---|---|---|---|
| Sheng et al. (2022) | 2022 | Text | Various | ✗ | N/A | ✗ | ✗ |
| Cheng et al. (2023) | 2023 | Text | Various | ✗ | N/A | ✗ | ✗ |
| Zhao et al. (2024) | 2024 | Text | Various | ✗ | N/A | ✗ | ✗ |
| Yan et al. (2023) | 2023 | Voice | Various | ✗ | N/A | ✗ | ✗ |
| Wan et al. (2024) | 2024 | Fed. Learn. | Various | ✗ | N/A | ✗ | ✗ |
| Le Roux et al. (2024) | 2024 | Image (Face) | Various | ✗ | N/A | ✗ | ✗ |
| Li et al. (2022) | 2022 | Image | Various | ✗ | N/A | ✗ | ✗ |
| Wu et al. (2022) | 2022 | Image | Various | ✓ | 5 | ✗ | ✓ |
| **Ours** | 2026 | Image | **Mitigation** | ✓ | **19** | ✓ | ✓ |

*Existing surveys and benchmarks:* Several surveys and benchmarks study backdoor attacks and defenses. Table 1 positions our work relative to the closest prior surveys and benchmarks. Most existing surveys take a broader view than ours, spanning multiple defensive tasks or application domains. This breadth is valuable, but it leaves less room for method-level analysis of mitigation methods in image classification. The two closest works are Li et al. (2022) and Wu et al. (2022). Li et al. (2022) surveys multiple classification tasks, but its image-classification analysis remains high level, with limited depth on per-method assumptions or optimization structure. Wu et al. (2022) primarily develops a benchmarking tool and, although it covers nine defensive methods, only five perform mitigation under the definition used here. Although Wu et al. (2022) has incorporated additional methods since publication, consistent large-scale benchmarking of these works has not, to our knowledge, been conducted. Two gaps therefore remain: (i) existing image-domain surveys do not provide per-method depth on the assumptions, optimization structure, and resulting limitations of mitigation approaches; and (ii) the impact of practical constraints such as limited mitigation-data availability is not systematically studied.

*Our contributions:* This work addresses these gaps. We focus exclusively on *mitigation* for image classification, distinguishing it from related defensive tasks such as backdoor detection, identification of poisoned training examples, and synthesis of backdoored inputs. Specifically:

- We provide a focused review of 23 prominent mitigation methods, grouped into pruning (metric-based, masking-based, additive) and fine-tuning (conventional, knowledge-distillation, synthesis-unlearn, adversarial) categories, and benchmark 19 of them in a unified evaluation. For each method we describe its motivating observation, optimization problem, key assumptions, and limitations, enabling a like-for-like comparison that prior surveys do not support.

- We conduct over 120,000 individual experiments spanning eight backdoor attacks, three datasets, four model architectures, three poisoning ratios, and three data-availability settings (2, 10, and 100 samples per class). To our knowledge, this is the largest consistent evaluation of mitigation methods to date.

- We aggregate our results to support the claim that, despite a steady stream of new methods, improvements over the seminal 2018 baselines FP and FT remain modest and inconsistent across settings (Section 6).

- Drawing on these findings, we identify common failure modes that recur across method categories, link them to the assumptions made by individual methods, and outline open challenges to guide future research.

## 2 Background and Preliminaries

In this section, we introduce the concepts and notation used throughout the survey.

Table 2: The list of common symbols.

| Symbol | Meaning |
|--------|---------|
| $\theta$ | Model parameters |
| $\varphi$ | Parameters in $\theta$ associated with feature extractor |
| $\omega$ | Parameters in $\theta$ associated with the linear classifier |
| $\phi$ | Filter matrix of a convolutional layer |
| $\xi$ | Parameter perturbation applied to $\theta$ |
| $\delta$ | Input perturbation applied to $x$ |
| $\epsilon$ | Perturbation budget |
| $\lambda$ | Loss hyperparameter |
| $p$ | Norm type used to define $\|\cdot\|_p$ |
| $\mathcal{X} \subseteq \mathbb{R}^{h \times w \times c}$ | Input space with $c$ channels, $w$ width and $h$ height |
| $x \in \mathcal{X}$ | Clean input image |
| $\hat{x} \in \mathcal{X}$ | Backdoored image |
| $\rho$ | Trigger pattern applied to $x$ to produce $\hat{x}$ |
| $\mathcal{M} = \{0, 1, \ldots, m\}$ | Label space |
| $y \in \{0, 1\}^m$ | Correct label associated with $x$ (One-hot encoded) |
| $\hat{y} \in \{0, 1\}^m$ | Target label associated with $\hat{x}$ (One-hot encoded) |
| $(x, y) \in \mathcal{D}_t$ | Training dataset |
| $(x, y) \in \mathcal{D}_c$ | Clean dataset |
| $(\hat{x}, \hat{y}) \in \mathcal{D}_b$ | Backdoor dataset |
| $(x, y) \in \mathcal{D}_m$ | Mitigation dataset |
| $(x, y) \in \mathcal{D}_v$ | Validation dataset |
| $z \in \mathbb{Q}^m$ | Logit output of a model |
| $a \in [0, 1]^m$ | Softmax output of a model |

## 2.1 Notation

Table 2 summarizes the common symbols used throughout the paper.

### 2.1.1 Cross-Entropy Loss

Let $a = f(x, \theta) \in \mathbb{R}^m$ be the softmax output of a network for input $x$ and parameters $\theta$, and let $y \in \{0, 1\}^m$ be the one-hot label. The sample-wise and dataset-wise cross-entropy losses are

$$\mathcal{L}_{\text{CE}}(x, y; \theta) = -y^\top \log a, \quad \mathcal{L}_{\text{CE}}(\mathcal{D}; \theta) = \frac{1}{|\mathcal{D}|} \sum_{(x,y) \in \mathcal{D}} \mathcal{L}_{\text{CE}}(x, y; \theta).$$

### 2.1.2 Model Training

A neural network is a parameterized function $f(\cdot, \theta)$ that is commonly trained by solving

$$\theta^\star = \arg\min_\theta \mathcal{L}_{\text{CE}}(\mathcal{D}_{\text{train}}, \theta), \tag{1}$$

typically with some variation of gradient descent. For image classification, $f$ is often a convolutional network that can be factorized into a feature extractor $\varphi$ and a linear classifier $\omega$:

$$f(x, \theta) = \omega\big(\varphi(x)\big), \quad \theta = \varphi \cup \omega, \ \varphi \cap \omega = \varnothing.$$

## 2.2 Backdoor Attacks

Most training-time backdoor attacks construct a poisoned training set $\widetilde{\mathcal{D}}_t$ by combining clean samples $\mathcal{D}_c$ with backdoor samples $\mathcal{D}_b$. A key design variable is the poisoning ratio, i.e., the fraction of training samples that are backdoored, which controls the trade-off between stealth and attack effectiveness. The clean dataset $\mathcal{D}_c$ contains unaltered inputs $x$ and labels $y$. The backdoor dataset $\mathcal{D}_b$ is generated by a backdoor function $B(x, y, \rho) \mapsto (\hat{x}, \hat{y})$, where $\rho$ denotes the trigger and $\hat{x}$ is the triggered input. In targeted attacks, which are

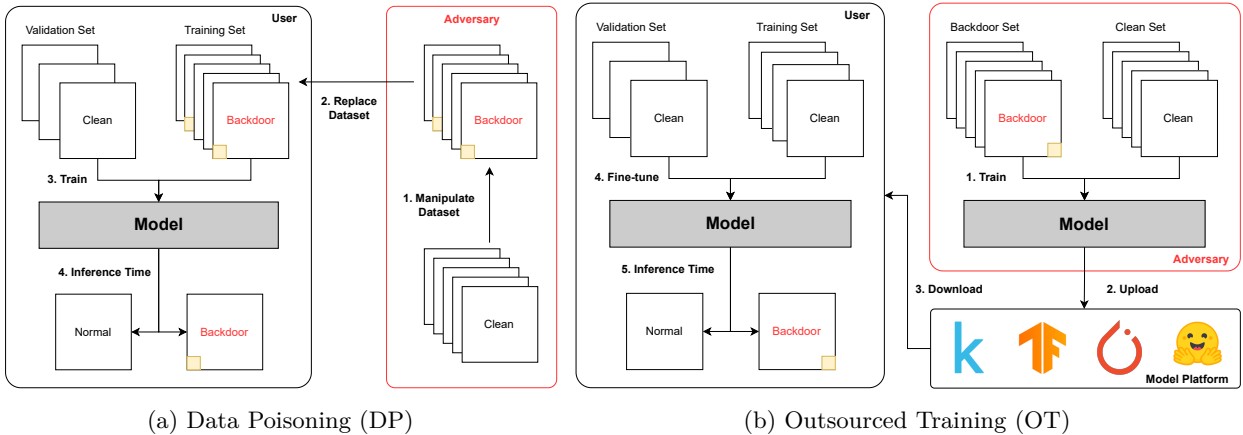

(a) Data Poisoning (DP)        (b) Outsourced Training (OT)

Figure 2: Threat models considered by existing backdoor attacks.

Table 3: Categorization of representative backdoor attacks based on key characteristics and the training procedure employed by the adversary. DP: Data Poisoning, OT: Outsourced Training.

| Ref | Name | Trigger Characteristics | | | Threat |
|---|---|---|---|---|---|
| | | Coverage | Consistency | Mode | Model |
| Gu et al. (2017) | BadNets | Local | Static | Replacement | DP |
| Chen et al. (2017) | Blended | Global | Static | Additive | DP |
| Barni et al. (2019) | Signal | Global | Static | Additive | DP |
| Zeng et al. (2021) | LF | Global | Static | Additive | DP |
| Li et al. (2021b) | SSBA | Global | Dynamic | Additive | DP |
| Nguyen & Tran (2020) | IAB | Local | Dynamic | Replacement | OT |
| Wang et al. (2022) | BPP | Global | Dynamic | Additive | OT |
| Nguyen & Tran (2021) | WaNet | Global | Dynamic | Warping | OT |
| Doan et al. (2021) | LIRA | Global | Dynamic | Additive | OT |

the most widely studied setting, $\hat{y}$ is assigned to a predefined target class. Alternative objectives, such as the all-target objective described in Zhao et al. (2020), have also been explored, but targeted attacks remain the dominant setting in the mitigation literature. Using $\widetilde{\mathcal{D}}_t$, the adversary optimizes $\theta$ so that the model classifies both $\mathcal{D}_c$ and $\mathcal{D}_b$ according to their intended labels. Some attacks, such as LIRA Doan et al. (2021), introduce specialized training procedures to strengthen the backdoor. These attack-training procedures are relevant background, but they fall outside the scope of our mitigation benchmark.

Current backdoor attacks are commonly analyzed under two threat models, as shown in Figure 2. The first, the *Data Poisoning* threat model, limits the adversary to modifying the training data, allowing them to replace $\mathcal{D}_t$ with $\widetilde{\mathcal{D}}_t$. The second, the stronger *Outsourced Training* threat model, gives the adversary control over the training process. In this setting, the adversary can not only substitute $\mathcal{D}_t$ with $\widetilde{\mathcal{D}}_t$ but also use an arbitrary, non-auditable training procedure. Using these threat models, along with additional characteristics introduced in studies such as Wu et al. (2022), we categorize prominent image-classification backdoor attacks in Table 3. Recent work continues to introduce more specialized attacks, including backdoor reactivation Zhu et al. (2024a) and sample-specific clean-label attribute triggers Zhu et al. (2025). We do not add such attacks to the benchmark because they introduce additional assumptions or attack mechanisms beyond the representative attack grid considered here. Instead, we focus on a widely used set of general-purpose attacks that spans trigger coverage, consistency, modification mode, and threat model, allowing us to assess mitigation robustness across a controlled but diverse range of attack settings.

### 2.2.1 Trigger Characteristics

Trigger characteristics pertain to the type of trigger $\rho$ used by the adversary, encompassing coverage, consistency and modification mode, as defined in Wu et al. (2022). Below, we briefly discuss these characteristics with reference to the works listed in Table 3.

**Coverage**  Trigger coverage refers to the extent of the input modification when $\rho$ is integrated into $x$ by $B$ Wu et al. (2022). In Figure 1, we provide examples of backdoor images generated by the considered attacks. Coverage is typically classified as global or local. Global coverage implies that $\rho$ affects a significant portion of $x$. For instance, the Blended attack Chen et al. (2017) alters $x$ by incorporating a trigger image (e.g., a picture of Hello Kitty) with a blending ratio that controls its transparency. In contrast, local triggers modify only a small portion of $x$. Among the attacks considered, BadNets Gu et al. (2017) and IAB Nguyen & Tran (2020) use local triggers. In the case of BadNets, a small $n \times n$ pixel pattern (e.g., a $3 \times 3$ white square) is inserted into images at a fixed position (e.g., in the lower-left corner).

**Consistency**  Trigger consistency refers to whether the same $\rho$ is used across $\mathcal{D}_b$. When $\rho$ is fixed, the attack is deemed static. For example, the BadNets attack inserts the same $n \times n$ pixel pattern into the same position in each image. However, the success of works such as Wang et al. (2019b) in identifying static triggers used by BadNets has led to the preference for dynamic triggers in recent works. More specifically, dynamic methods often make $\rho$ dependent on $x$. For example, IAB generates a unique $\rho$ for each $(x, y) \in \mathcal{D}_c$. Unlike other methods such as SSBA, BPP, and LIRA, which synthesize barely perceptible patterns, IAB generates patterns similar to those used in BadNets. Moreover, the triggers generated by IAB for different images are designed to be non-reusable, meaning the trigger for $x$ does not work for $x'$.

**Mode**  Trigger modification mode refers to how $\rho$ is applied to $x$. The two most common modes are additive and replacement. Replacement attacks substitute part of $x$ with $\rho$. For example, BadNets replaces the pixels in the $n \times n$ region of $x$ that overlaps with $\rho$. In contrast, additive attacks add $\rho$ to $x$ (i.e., $\hat{x} = x + \rho$). WaNet is an exception because its elastic image-warping transformation is neither purely additive nor purely replacement-based.

### 2.3 Backdoor Mitigation

Defenses against backdoor attacks address several distinct sub-problems. Some methods identify poisoned samples within $\widetilde{\mathcal{D}}_t$ Gao et al. (2019); Hou et al. (2025). Others synthesize candidate triggers or backdoored inputs from clean data and model parameters Liu et al. (2019). More recent work, such as Chen et al. (2025), learns an input transformation that suppresses the backdoor effect without modifying the underlying model. These are valuable defensive directions, but they require different assumptions and evaluation protocols from the mitigation setting studied here. We focus on backdoor mitigation, where the objective is to modify the compromised model itself so that backdoor behavior is removed while clean-task performance is preserved Liu et al. (2018).

Although several works address backdoor mitigation, they often differ in the threat model assumed. To ensure a fair comparison, we focus on proposals whose assumptions are compatible with the strongest adversarial scenario considered here: the outsourced-training threat model outlined in Section 2.2. In this setting, the defender, who performs the mitigation, has access to $\theta$ and a small set of clean mitigation data $\mathcal{D}_m$. Crucially, the defender does not have access to any backdoor data $\mathcal{D}_b$. In addition, our review prioritizes prominent works from established peer-reviewed venues, particularly CORE[1] A/A* conferences and SJR[2] Q1 journals, while also including closely related preprints when they are directly relevant to the evaluated method set.

In the following sections, we present a comprehensive analysis of existing backdoor mitigation approaches within the image classification domain. For each work, we summarize the methodology and highlight the assumptions that underlie the proposed mitigation strategy. We broadly group works into two main categories, pruning and fine-tuning, while also providing more granular sub-typing.

---

[1] https://www.core.edu.au/icore-portal
[2] https://www.scimagojr.com/journalrank.php

Table 4: Categorization of the surveyed backdoor mitigation approaches.

| Ref | Name | Category | Type |
|---|---|---|---|
| Liu et al. (2018) | FP | | Metric |
| Zheng et al. (2022a) | BNP | | |
| Zheng et al. (2022b) | CLP | | |
| Wu & Wang (2021) | ANP | | Masking |
| Chai & Chen (2022) | AWM | | |
| Li et al. (2023) | RNP | Pruning | |
| Huang & Bu (2023) | FMP | | |
| Karim et al. (2024) | NFT | | |
| Wang et al. (2023) | MM-BD | | Additive |
| Cheng et al. (2024) | UNIT | | |
| Zhu et al. (2024b) | NPD | | |
| Wang et al. (2019a) | NC* | | Synthesis Unlearn |
| Qiao et al. (2019) | MESA | | Synthesis Unlearn |
| Liu et al. (2022) | BAERASER | | |
| Min et al. (2024) | FST | | Traditional |
| Zhu et al. (2023) | FT-SAM | | |
| Xu et al. (2024) | BTI-DBF | Fine-Tuning | |
| Li et al. (2021a) | NAD | | Knowledge Distillation |
| Pang et al. (2023) | BCU | | |
| Zeng et al. (2022) | i-BAU | | Adversarial Training |
| Mu et al. (2023) | PBE | | |
| Wei et al. (2023) | SAU | | |

*NC shares the trigger-synthesis step used by synthesis-unlearn methods, but its final mitigation step is pruning.

# 3   Model Pruning

When backdoor attacks were first introduced for image classification, Gu et al. (2017) hypothesized that compromised models learn two distinct sets of features. Under this view, some model components, such as convolutional or dense layers, become associated with either the clean task or the backdoor task. Consequently, when the model is presented with $\hat{x}$, the components contributing to the backdoor task result in its classification as $\hat{y}$ rather than $y$.

Historically, model pruning has been employed to identify and remove redundant model components, thereby improving inference efficiency He & Xiao (2023). Building on the hypothesis that model components specialize, several studies have explored the application of model pruning for backdoor mitigation. Pruning-based approaches aim to identify and eliminate model components associated with backdoor behavior, while keeping those responsible for the clean task. To achieve this, various strategies have been adopted, including metric-based, masking-based, and additive techniques. In this section, we provide a comprehensive review of these subcategories, along with a comparative analysis of the approaches within them.

## 3.1   Metric-based Pruning

Works adopting a metric-based approach aim to directly quantify the contribution of each model component to the backdoor task. By applying a defined metric, these works distinguish between clean and backdoor components based on their respective metric values.

### 3.1.1   FP

The work of Liu et al. (2018) is considered a pioneering effort in backdoor mitigation. In their initial investigation, Liu et al. (2018) compares the channel-wise average activation produced by $\mathcal{D}_c$ and $\mathcal{D}_b$ in the final convolutional layer. Their analysis reveals that backdoor components, specifically a small subset of filters within this layer, are only activated by $\mathcal{D}_b$.

Building on this observation, Liu et al. (2018) proposes pruning the final convolutional layer based on the average channel-wise activation given $\mathcal{D}_m$, the mitigation dataset available to the defender. The approach

involves iteratively pruning filters with the lowest average activation until the accuracy on a validation dataset $\mathcal{D}_v$, which is segmented from $\mathcal{D}_m$ before pruning, falls below a defined threshold. After pruning, Liu et al. (2018) fine-tune $\theta$ using $\mathcal{D}_m$ to recover lost clean accuracy.

### 3.1.2 BNP

Similar to FP Liu et al. (2018), Zheng et al. (2022a) examine the activation values of backdoored models. However, Zheng et al. (2022a) compare the pre-activation distribution of clean and backdoor components. Pre-activation refers to the activation values before any non-linear transformation, e.g., ReLU, is applied. Their analysis reveals that clean components typically follow a unimodal distribution, while backdoor components exhibit a bimodal distribution (see (Zheng et al., 2022a, Fig. 2(a)-(b))).

As an extension of the above analysis, Zheng et al. (2022a) compare the batch normalization (BN) statistics ($\mu_{bn}$ and $\sigma_{bn}$) tracked in the corresponding BN layer with the pre-activation statistics of $\mathcal{D}_c$ ($\mu_c$ and $\sigma_c$). Their comparison shows that $\mu_{bn}$ and $\sigma_{bn}$ are biased relative to $\mu_c$ and $\sigma_c$, a consequence of $\mu_{bn}$ and $\sigma_{bn}$ stemming from the bimodal input distribution produced by $\widetilde{\mathcal{D}}_t$ (see (Zheng et al., 2022a, Fig. 2(c)-(d))).

To leverage this characteristic, Zheng et al. (2022a) calculate the Kullback-Leibler (KL) divergence between $\mathcal{N}(\mu_{bn}, \sigma_{bn})$ and $\mathcal{N}(\mu_m, \sigma_m)$. The KL divergence is calculated for each filter within each convolutional layer. The set of KL-divergence values for the $l^{\text{th}}$ layer, denoted as $K_l = \{k_1, k_2 \ldots, k_n\}$, is then used to determine a layer-specific pruning threshold $\tau_l$, calculated as

$$\tau_l = \bar{K}_l + \lambda s_l, \tag{2}$$

where $\bar{K}_l$ and $s_l$ are the mean and standard deviation of $K_l$, and $\lambda$ is a hyperparameter selected by the defender. The filters are then pruned based on $\tau_l$. However, it is important to note that BNP assumes that a subset of filters exhibits biased BN statistics compared to the pre-activation statistics of $\mathcal{D}_c$. While this characteristic is demonstrated for a single filter in (Zheng et al., 2022a, Fig. 2(c)-(d)), the distribution of $K_l$ across all layers is not provided, leaving limited evidence to support the assertion that filters in each layer can be distinctly separated using the proposed metric. Moreover, given the inherent complexity and high non-linearity in modern classification architectures, the assumption that the pre-activation distribution of all layers is Gaussian is also unlikely to hold.

### 3.1.3 CLP

Unlike other metric-based methods, Zheng et al. (2022b) propose using the *Lipschitz* constant associated with each filter matrix $\phi$ to guide pruning. For the $i^{\text{th}}$ filter in the $l^{\text{th}}$ layer, the upper bound channel Lipschitz constant (UCLC) of $\phi_{l,i}$ is estimated as the largest singular value from the spectral decomposition of $\phi_{l,i}$. Note that $\phi_{l,i}$ is reshaped such that $\phi_{l,i} \in \mathbb{R}^{c \times (hw)}$, where $c$, $h$, and $w$ represent the channel, height, and width dimensions of $\phi_{l,i}$.

Operating under the hypothesis that backdoor components exhibit distinct activation patterns for $\mathcal{D}_c$ and $\mathcal{D}_b$, Zheng et al. (2022b) argue that the UCLC can effectively quantify this difference without needing access to either dataset. To validate this idea, Zheng et al. (2022b) introduces the trigger activation change (TAC), which quantifies the average activation difference between $\mathcal{D}_c$ and $\mathcal{D}_b$ for the $i^{\text{th}}$ filter in the $l^{\text{th}}$ layer, as

$$\frac{1}{|\mathcal{D}_c|} \sum_{(x,y)\in\mathcal{D}_c, (\hat{x},\hat{y})\in\mathcal{D}_b} \|f_{l,i}(x,\theta) - f_{l,i}(\hat{x},\theta)\|_2, \tag{3}$$

where $(x, y) \in \mathcal{D}_c, (\hat{x}, \hat{y}) \in \mathcal{D}_b$ is a clean and backdoor image pair. By plotting UCLC and TAC against each other, Zheng et al. (2022b) demonstrate a strong positive correlation between the two metrics (see (Zheng et al., 2022b, Fig. 3)). They subsequently suggest that UCLC can reliably quantify the sensitivity of each filter to $\rho$, allowing for a distinction between clean and backdoor components. For model pruning, Zheng et al. (2022b) adopts the same layer-based thresholding approach as BNP Zheng et al. (2022a).

Although the proposed metric is well correlated with TAC in the initial analysis of Zheng et al. (2022b), its generalizability across architectures is not well demonstrated. Because UCLC is an upper bound, its sensitivity to changes in dataset, architecture, and attack type remains unclear.

### 3.1.4 FMP

Distinct from other metric-based pruning methods, FMP Huang & Bu (2023) directly estimates the contribution of individual convolutional filters to classification behavior. The key idea is to measure how sensitive each filter's feature map is to input perturbations that maximize the difference between clean and perturbed feature maps.

For the $i^{\text{th}}$ filter in the $l^{\text{th}}$ layer, FMP constructs adversarially perturbed inputs by solving

$$\max_{\{\delta_{l,i}^{(x)}\}_{(x,y)\in\mathcal{D}_m}} \sum_{(x,y)\in\mathcal{D}_m} \|f_{l,i}(x,\theta) - f_{l,i}(x + \delta_{l,i}^{(x)}, \theta)\|_2, \tag{4}$$

where $\delta_{l,i}^{(x)}$ is the filter-specific perturbation for sample $x$, and $f_{l,i}(\cdot,\theta)$ denotes the feature map produced by the $i^{\text{th}}$ filter in layer $l$. The resulting perturbed mitigation set can be written as

$$\widetilde{\mathcal{D}}_m = \{(\tilde{x}, y, x) \mid \tilde{x} = x + \delta_{l,i}^{(x)}, \ (x,y) \in \mathcal{D}_m\}.$$

After estimating these perturbations, FMP ranks filters by the induced feature-map change and prunes the top $p$ fraction of filters judged most sensitive.

A critical assumption is that filters associated with backdoor behavior can be identified independently through their sensitivity to such perturbations. If the backdoor effect emerges from the joint activation of multiple filters, rather than from individually separable filters, FMP may underestimate the importance of jointly responsible components.

## 3.2 Masking-based Pruning

Beyond metric-based approaches, some pruning methods focus on learning a parameter mask $\mathbf{m}$ that, when applied to $\theta$, effectively removes the backdoor behavior. These methods formulate an objective function and use optimization techniques to find an optimal mask $\mathbf{m}$. The mask is applied using the Hadamard (element-wise) product as $\mathbf{m} \odot \theta$. Hence, when an entry of $\mathbf{m}$ is zero, the corresponding entry in $\theta$ is pruned. Additionally, since $\mathbf{m}$ is used to mask convolutional filters, a mask value is learned for each filter in each layer.

### 3.2.1 ANP

In Wu & Wang (2021), the authors initially frame the backdoor mitigation task as a masking problem, where they examine the impact of a perturbation $\xi$ applied to $\theta$ on the classification error over $\mathcal{D}_m$ for both clean and backdoored models. The perturbation set $\xi$ consists of two subsets: $\xi_w$, applied to the weights ($w$) in $\theta$, and $\xi_b$, applied to the biases ($b$) in $\theta$. Therefore, $\xi = \xi_w \cup \xi_b$ and $\theta = w \cup b$. To determine $\xi$, the following optimization problem is solved

$$\max_{\xi_w, \xi_b \in [-\epsilon, \epsilon]} \mathcal{L}_{\text{CE}}(\mathcal{D}_m, [(1 + \xi_w) \odot w \ \cup \ (1 + \xi_b) \odot b]), \tag{5}$$

where $\epsilon$ constrains the values of $\xi$. In Wu & Wang (2021), it is shown that a backdoored model exhibits higher classification error (see (Wu & Wang, 2021, Fig. 1(a))). It is also observed that classification errors made by the backdoored model are biased toward the target class (see (Wu & Wang, 2021, Fig. 1(b))). Based on these results, Wu & Wang (2021) hypothesize that $\xi$ targets backdoor components of the model. To leverage this characteristic, they propose solving the following minimax optimization problem

$$\min_{\mathbf{m} \in [0,1]} \left\{ \lambda \mathcal{L}_{\text{CE}}(\mathcal{D}_m, [(\mathbf{m} + \xi_w) \odot w, b]) + \max_{\substack{\xi_w, \xi_b \\ \in [-\epsilon, \epsilon]}} (1 - \lambda) \mathcal{L}_{\text{CE}}(\mathcal{D}_m, [(\mathbf{m} + \xi_w) \odot w, (1 + \xi_b) \odot b]) \right\} \tag{6}$$

where $\lambda \in [0,1]$ is a trade-off coefficient chosen by the defender. Solving equation 6 yields a perturbation $\xi$ that maximizes the classification error of $\mathcal{D}_m$. At the same time, the outer minimization results in a mask $\mathbf{m}$ that minimizes the classification error of $\mathcal{D}_m$ given $\xi$. In Wu & Wang (2021), they alternate between solving the inner and outer sub-problems multiple times. Once a solution for $\mathbf{m}$ is found, it is binarized using a threshold value or a fixed pruning fraction.

While the proposed method is unique, the weak direct relationship between successive inner-maximization steps has the potential to limit ANP's effectiveness. Given that the inner step applies the learned mask to the model weights, repeated execution creates a weak coupling between the past and future iterations.

### 3.2.2  AWM

The robustness of ANP Wu & Wang (2021) in limited data settings is analyzed in Chai & Chen (2022). The analysis findings indicate that ANP becomes ineffective when fewer than 100 data samples are available (see (Chai & Chen, 2022, Fig. 1)). To overcome the impact of limited data on the performance of weight masking, Chai & Chen (2022) propose applying perturbations to inputs rather than to $\theta$. They redesign the inner maximization problem to identify an input perturbation $\delta$ that maximizes classification error when applied to $\mathcal{D}_m$. Given $\widetilde{\mathcal{D}}_m = \{(\tilde{x}, y) \mid \tilde{x} = x + \delta \mid (x,y) \in \mathcal{D}_m\}$, the inner maximization problem is expressed as

$$\max_{\|\delta\|_1 \leq \epsilon} \mathcal{L}_{\mathrm{CE}}(\widetilde{\mathcal{D}}_m, \theta), \tag{7}$$

where $\|\delta\|_1$ is bound by $\epsilon$. Notably, the same $\delta$ is applied to all elements of $\mathcal{D}_m$. Thus, Chai & Chen (2022) solve the following minimax optimization problem

$$\min_{\mathbf{m} \in [0,1]} \left\{ \lambda_1 \mathcal{L}_{\mathrm{CE}}(\mathcal{D}_m, \mathbf{m} \odot \theta) + \lambda_2 \max_{\|\delta\|_1 \leq \epsilon} \left[ \mathcal{L}_{\mathrm{CE}}(\widetilde{\mathcal{D}}_m, \mathbf{m} \odot \theta) \right] + \lambda_3 \|\mathbf{m}\|_1 \right\}, \tag{8}$$

where $\lambda_1$, $\lambda_2$, and $\lambda_3$ are hyperparameters that balance the influence of the three loss terms, and $\|\mathbf{m}\|_1$ serves an additional regularization term to encourage sparsity. Note that Chai & Chen (2022) omit the final binarization step employed in ANP, retaining $\mathbf{m}$ as a soft mask.

Promoting sparsity in $\mathbf{m}$ leads to significant pruning of $\theta$, which is expected to result in low bias and high variance given $\mathcal{D}_m$. Moreover, this additional term is not normalized to account for the size of $\mathbf{m}$. Therefore, identifying optimal values for $\lambda_1$, $\lambda_2$, and $\lambda_3$ that yield consistent performance across different model architectures is challenging. Moreover, this design also suffers from the same weak coupling issue discussed above.

### 3.2.3  RNP

Distinct from ANP Wu & Wang (2021) and AWM Chai & Chen (2022), Li et al. (2023) introduce a unique unlearning strategy. Rather than learning $\xi$ or $\delta$, Li et al. (2023) first *unlearn* the clean task by solving the following optimization problem

$$\max_{\theta} \mathcal{L}_{\mathrm{CE}}(\mathcal{D}_m, \theta). \tag{9}$$

The resulting set of unlearned parameters is denoted as $\hat{\theta}$. In Li et al. (2023), it is argued that this process leads to unlearning of the model's clean components while preserving the backdoor. Moreover, they suggest that the unlearned model exhibits biased misclassification toward the backdoor target.

Using $\hat{\theta}$, Li et al. (2023) then proceed to learn $\mathbf{m}$ by solving the following optimization problem

$$\min_{\mathbf{m} \in [0,1]} \mathcal{L}_{\mathrm{CE}}(\mathcal{D}_m, \mathbf{m} \odot \hat{\theta}). \tag{10}$$

The authors of Li et al. (2023) assert that this recovery procedure can differentiate between clean and backdoor filters, given that $\hat{\theta}$ exhibits biased misclassification toward the backdoor target. Specifically, they

suggest that this procedure removes backdoor filters by setting their corresponding elements in $\mathbf{m}$ to 0. Importantly, once an optimal solution to $\mathbf{m}$ is obtained, it is applied to the original parameters, $\theta$. In a manner similar to ANP, Li et al. (2023) binarize $\mathbf{m}$ using a threshold value or a fixed pruning ratio.

### 3.2.4 NFT

Karim et al. (2024) replace the adversarial trigger synthesis used in prior backdoor mitigation methods, such as AWM, with the MixUp Zhang et al. (2018) data augmentation strategy. Given two clean datapoints $i$ and $j$ from $\mathcal{D}_c$, MixUp forms augmented samples

$$\tilde{x}_{i,j} = \gamma x_i + (1-\gamma)x_j, \quad \tilde{y}_{i,j} = \gamma y_i + (1-\gamma)y_j, \tag{11}$$

where $\gamma \in [0,1]$ is drawn from a Beta distribution $\beta_{\alpha,\beta}$ and $y_i, y_j$ are one-hot vectors. Using this data augmentation strategy, NFT minimizes the MixUp objective

$$\mathcal{L}_{\text{MixUp}}(\mathcal{D}_c, \theta, \mathbf{m}) = \frac{1}{|\mathcal{D}_c|} \sum_{(x_i,y_i)\in\mathcal{D}_c} \mathbb{E}_{\gamma\sim\beta_{\alpha,\beta}} \left[ \mathcal{L}_{\text{CE}}\big(\tilde{x}_{i,j}, \tilde{y}_{i,j}; \theta \odot \mathbf{m}\big) \right], \tag{12}$$

$$\min_{\mathbf{m}} \ \mathcal{L}_{\text{MixUp}}(\mathcal{D}_c, \theta, \mathbf{m}) + \lambda\|\mathbf{m}\|_1, \quad \lambda = \frac{5 \times 10^{-4}}{n_c}, \tag{13}$$

where $n_c$ is the number of classes. Moreover, they set a dynamic layer-wise lower bound on $\mathbf{m}$, which decreases with depth to encourage pruning of deeper layers.

Theoretical analysis in Karim et al. (2024) shows that $\mathcal{L}_{\text{MixUp}}$ serves as an upper bound on the ideal purification loss

$$\mathcal{L}^*_{\text{purify}}(\mathcal{D}_b, \theta, \mathbf{m}) = \frac{1}{|\mathcal{D}_b|} \sum_{(x,y)\in\mathcal{D}_b} \mathcal{L}_{\text{CE}}\big(\hat{x}, y; \theta \odot \mathbf{m}\big), \tag{14}$$

where $\mathcal{D}_b$ is the inaccessible backdoor dataset and $y$ is the original label associated with $\hat{x}$.

A critical limitation of the paper's theoretical argument is its assumption about the nature of the backdoor data. The original MixUp proof posits that an effective defense rests on the premise that a backdoor trigger can be modeled as a small-magnitude perturbation of a clean image. This allows the ideal loss on a triggered sample to be mathematically bounded by the loss on samples created through MixUp. However, this assumption does not hold for many common backdoor attacks. Triggers like BadNets or other distinct colored regions are not small, distributed noise. Instead, they are large, localized changes to the input. In these cases, the backdoor data point does not necessarily lie on the manifold between pairs of clean data points in the way the proof assumes.

## 3.3 Additive Pruning

The pruning methods discussed thus far either directly prune or mask existing model components. In contrast, Wang et al. (2023), Cheng et al. (2024) and Zhu et al. (2024b) introduce additional model components that integrate with the existing structure. These additional components function as quasi-filters, targeting the removal of backdoor tasks through a mechanism akin to pruning. Consequently, we categorize these approaches loosely under the umbrella of pruning.

### 3.3.1 MM-BD and UNIT

In Wang et al. (2023) and Cheng et al. (2024), the task of backdoor mitigation is framed as a bounding problem. Initially, both works observe that backdoor samples tend to trigger large activations, with Wang et al. (2023) showing that this results in unusually high decision-making confidence (see (Wang et al., 2023, Fig. 9)). To quantify this difference in confidence, Wang et al. (2023) calculate the maximum margin statistic of $x$ given $y$ as

$$\mathcal{G}(x, y, \theta) = s_y(a) - \max_{k \in \mathcal{M}\setminus y} s_k(a), \tag{15}$$

where $s_n$ selects the $n$th logit from the softmax output $a$ given $x$, and $\mathcal{M}$ represents the set of possible labels. Their analysis reveals that backdoor samples exhibit significantly larger confidence compared to clean samples (see (Wang et al., 2023, Fig. 4(a))). The authors conjecture that this is due to the abnormally large activations influencing the model's decision-making.

To counteract the impact of these large activations have on decision-making, Wang et al. (2023) propose learning a set of upper-bound values $\mathbf{B} = \{b_1, \dots, b_L\}$ for each non-linear activation layer, such as ReLU. These bounds are learned in a channel-wise manner and used during the forward pass to constrain activation range within each channel. To learn $\mathbf{B}$, the following optimization problem is solved

$$\min_{\mathbf{B}} \frac{1}{|\mathcal{D}_m| \times |\mathcal{M}|} \sum_{(x,y) \in \mathcal{D}_m} [f(x, \theta_{\mathbf{B}}) - f(x, \theta)]^2 + \lambda \|\mathbf{B}\|_2, \tag{16}$$

where $\theta_{\mathbf{B}}$ represents the model parameters $\theta$ combined with the learned bounding values $\mathbf{B}$. The objective is to find the bounding values $\mathbf{B}$ that minimally impact the classification of $\mathcal{D}_m$.

Similar to Wang et al. (2023), Cheng et al. (2024) find the minimum layer-wise upper bounds $\mathbf{B}$ that do not impact the classification of $\mathcal{D}_m$. The upper bound $b_i$ is initialized as $b_i = \mu_i + 4\sigma_i$, where $\mu_i$ and $\sigma_i$ are the mean and standard deviation of the activation distribution produced by $\mathcal{D}_m$ within layer $i$. They then solve the following optimization problem

$$\min_{\mathbf{B}} \mathcal{L}_{\mathrm{CE}}(\mathcal{D}_m, \theta_{\mathbf{B}}) + \lambda \|\mathbf{B}\|_1, \tag{17}$$

Similar to AWM, the regularization term used by both approaches is not normalized to account for variations in model architectures. In addition, this term does not adjust for differences in the scale of activation values across layers. Variations in activation scale are likely to directly affect the values of $\mathbf{B}$ and influence the choice of $\lambda$. Another critical assumption made is that setting an upper bound for each channel is sufficient to restore correct classification of $\mathcal{D}_b$, with both proposals only reporting if ASR is reduced.

### 3.3.2 NPD

Unlike MM-BD Wang et al. (2023) and UNIT Cheng et al. (2024), Zhu et al. (2024b) approaches backdoor mitigation from a more traditional pruning perspective. To implement pruning, Zhu et al. (2024b) introduces a 1x1 convolutional layer into the model, suggesting that this layer is added close to the final layer of the feature extraction. Referred to as a polarizer, this layer learns a set of parameters $\boldsymbol{w}$ that filter channels associated with the backdoor task. The 1x1 convolutional layer maintains the same number of channels as the preceding layer, and therefore scales each output channel by the corresponding value in $\boldsymbol{w}$. The augmented model $f_{w,\theta}$ is parameterized by $\theta$ and $\mathbf{w}$. However, $\theta$ remains fixed throughout and therefore is excluded from subsequent equations.

Similar to AWM Chai & Chen (2022), Zhu et al. (2024b) first approximates the backdoor trigger $\rho$ as an input perturbation $\delta$. However, unlike AWM, Zhu et al. (2024b) models the trigger distribution in a sample-specific manner, learning a distinct value of $\delta$ for each $(x, y) \in \mathcal{D}_m$. Given $x$, $\delta$ is learned by solving the following optimization problem

$$\min_{\|\delta\|_p \leq \epsilon} \mathcal{L}_{\mathrm{CE}}(x + \delta, \tilde{y}, \mathbf{w}), \tag{18}$$

where $\tilde{y}$ is an estimate of the target class, for which Zhu et al. (2024b) provide several heuristics. Under the assumption that the defender does not know the backdoor target, Zhu et al. (2024b) suggests using the second-largest logit for $x$. However, Zhu et al. (2024b) does not quantitatively validate how frequently the second-largest logit corresponds to the backdoor target. Using $\widetilde{\mathcal{D}}_m = \{(\tilde{x}, \tilde{y}, x, y) \mid \tilde{x} = x + \delta \mid (x, y) \in \mathcal{D}_m\}$, they solve the following optimization problem

$$\mathcal{L}_{\mathrm{ASR}}(\tilde{x}, \tilde{y}, \mathbf{w}) = -\log(1 - s_{\tilde{y}}(\tilde{a})), \tag{19}$$

$$\mathcal{L}_{\text{BCE}}(\tilde{x}, y, \mathbf{w}) = -\log(s_y(\tilde{a})) - \log(1 - \max_{k \neq y} s_k(\tilde{a})), \tag{20}$$

$$\min_{\mathbf{w}} \left\{ \frac{1}{|\widetilde{\mathcal{D}}_m|} \sum_{(\tilde{x}, \tilde{y}, x, y) \in \widetilde{\mathcal{D}}_m} \lambda_1 \mathcal{L}_{\text{CE}}(x, y, \mathbf{w}) + \lambda_2 \mathcal{L}_{\text{ASR}}(\tilde{x}, \tilde{y}, \mathbf{w}) + \lambda_3 \mathcal{L}_{\text{BCE}}(\tilde{x}, y, \mathbf{w}) \right\}, \tag{21}$$

where $\tilde{a} = f(\tilde{x}, \theta)$, and $\lambda_1$, $\lambda_2$ and $\lambda_3$ are hyperparameters that control the influence of each term. This design optimizes $\mathbf{w}$ to alleviate the impact of $\delta$ when applied to $\mathcal{D}_m$. The term $\mathcal{L}_{\text{ASR}}$ penalizes the classification of $\tilde{x}$ as $\tilde{y}$, while $\mathcal{L}_{\text{BCE}}$, similar to the margin statistic proposed in Wang et al. (2023), encourages confident classification of $\tilde{x}$ as $y$, its correct label. Finally, the loose coupling issue associated with ANP and AWM is not resolved in NPD, as the polarizer is used when estimating $\delta$.

### 3.4 Summary

Model pruning is an important strategy in mitigating backdoor attacks, building on the hypothesis that neural networks can be decomposed into distinct components responsible for either clean or backdoor tasks. By selectively pruning the components related to backdoor behavior, researchers aim to restore model integrity while preserving performance on the original task.

Metric-based pruning approaches quantify each model component's contribution to backdoor behavior through various metrics. Approaches like FP remove filters based on their activation patterns, while BNP uses distribution statistics to identify backdoor components. CLP's Lipschitz-based approach uses approximations of the channel Lipschitz constants to guide pruning decisions. Masking-based pruning techniques optimize a parameter mask to remove backdoor functionality. Approaches such as ANP, AWM, RNP and NFT employ optimization frameworks to iteratively refine the mask, targeting backdoor components while retaining model accuracy. Lastly, additive pruning approaches introduce new components into the network, functioning as filters that mitigate backdoor influences without directly removing existing filters.

## 4 Fine-Tuning

An alternative approach to model pruning for backdoor mitigation is fine-tuning. Instead of removing a subset of $\theta$, fine-tuning methods adjust the values of $\theta$ to eliminate the backdoor. A key aspect of these methods is the objective function, which typically incorporates one or more carefully designed regularization terms. These terms are usually selected based on specific insights gained from preliminary investigations. In this section, we review the most prominent fine-tuning approaches, categorized into distinct subgroups based on their unique methodologies.

### 4.1 Conventional Fine-Tuning

The first subcategory of fine-tuning methods follows a more conventional approach. Here, conventional refers to the proposed optimization problem being closely aligned with equation 1.

#### 4.1.1 FST

In their preliminary investigation, Min et al. (2024) evaluate the effectiveness of minimization equation 1 given $\mathcal{D}_m$ for mitigating backdoor tasks. Their analysis decomposes $\theta$ into $\varphi$, parameters associated with the feature extractor, and $\omega$, parameters associated with the linear classifier. They tested fine-tuning various combinations of $\varphi$ and $\omega$, concluding that the minimization equation 1 for any combination of these parameters was largely ineffective in removing backdoors (see (Min et al., 2024, Table 1)).

Building on these findings, Min et al. (2024) propose reinitializing $\omega$ by assigning new random values to $\omega$ before jointly fine-tuning both $\varphi$ and $\omega$. To enhance this process, they introduce a regularization term that

encourages divergence between the original and new values of $\omega$, referred to as $\hat{\omega}$ and $\omega$, respectively. This leads to minimizing the following objective function

$$\min_{\theta} \mathcal{L}_{\text{CE}}(\mathcal{D}_m, \theta) + \lambda \omega^{\mathsf{T}} \hat{\omega}, \quad \text{s.t. } \|\omega\|_2 = \|\hat{\omega}\|_2, \tag{22}$$

where $\lambda$ is a hyperparameter controlling the influence of the regularization term. According to Min et al. (2024), $\omega^{\mathsf{T}} \hat{\omega}$ discourages $\omega$ from learning the same relationships between the penultimate set of features and class labels. The constraint $\|\omega\|_2 = \|\hat{\omega}\|_2$ is applied to minimize the impact of $\omega^{\mathsf{T}} \hat{\omega}$ on the overall loss during fine-tuning. The norm constraint prevents this regularization term from dominating the cross-entropy loss solely through changes in weight magnitude.

### 4.1.2 FT-SAM

In addition, Zhu et al. (2023) investigate the effectiveness of traditional fine-tuning in mitigating backdoors. Similar to FST Min et al. (2024), they find it ineffective. Their analysis of the $\omega$ norms, the magnitudes of each parameter, revealed minimal changes following fine-tuning (see (Zhu et al., 2023, Fig. 2)). They hypothesize that fine-tuning fails because it is unable to escape the local minima it finds, allowing the backdoor to persist. Additionally, Zhu et al. (2023) demonstrates that $\omega$ norms are positively correlated with TAC [see equation 3], a metric previously used in CLP Zheng et al. (2022b) to quantify activation difference between $\mathcal{D}_c$ and $\mathcal{D}_b$.

Inspired by sharpness-aware minimization (SAM) methods Foret et al. (2020), Zhu et al. (2023) propose a minimax optimization method designed to escape sharp local minima. Similar to ANP Wu & Wang (2021), the inner maximization seeks an $\ell_2$-bounded perturbation $\xi$ that maximizes the classification loss for $\mathcal{D}_m$. However, rather than identifying a weight mask $\mathbf{m}$, the outer minimization step updates $\theta$ directly. This leads to solving the following optimization problem

$$\min_{\theta} \max_{\left\|\mathbf{T}_{\theta}^{-1} \xi\right\|_2 \leq \epsilon} \mathcal{L}_{\text{CE}}(\mathcal{D}_m, \theta + \xi), \tag{23}$$

where $\mathbf{T}_{\theta} = \text{diag}\left(|\theta_1|, \ldots, |\theta_L|\right)$, with $\theta_i$ being the $i^{\text{th}}$ parameter in $\theta$ and $\epsilon$ serving as a hyperparameter controlling the perturbation budget. The traditional $\ell_2$-bounding constraint is modified to $\|\mathbf{T}_{\theta}^{-1} \xi\|_2 \leq \epsilon$, allowing larger perturbations to be applied to elements of $\theta$ with larger norms, as their corresponding component in $\mathbf{T}_{\theta}^{-1}$ approaches zero. This evaluates the stability of $\theta$, quantified as $\mathcal{L}_{\text{CE}}$ on $\mathcal{D}_m$, when perturbed by $\xi$.

The perturbation constraint $\|\mathbf{T}_{\theta}^{-1} \xi\|_2 \leq \epsilon$ used by Zhu et al. (2023) is applied during each gradient-descent step. This implies that $\theta$ can drift significantly from its initial values after several steps. Moreover, since the estimation of $\xi$ relies on $\mathcal{D}_m$, the update can have high estimation variance.

### 4.2 Knowledge Distillation

Inspired by its success in other learning settings, two proposals explore how knowledge distillation (KD) can be leveraged for backdoor mitigation. Traditionally used to *transfer knowledge* from larger to smaller models, KD has been effectively applied in tasks such as image classification Gou et al. (2021). In the context of backdoor mitigation, the distillation process is reframed as a *knowledge filtering* task. Instead of transferring all knowledge from the original model, the goal of *knowledge filtering* is to distill only the information relevant to the clean task, thereby eliminating the backdoor-related information.

A common approach to KD involves a teacher-student architecture. In typical applications, the teacher model is a larger, more capable model whose knowledge is transferred to a smaller student model. However, in the case of backdoor mitigation, access to a non-backdoored teacher model is not possible. Subsequently, approaches that employ this method must overcome this challenge.

### 4.2.1 NAD

To implement KD, Li et al. (2021a) introduce a new attention-based method. Rather than relying on feature maps (i.e., intermediate activation outputs) to perform KD, Li et al. (2021a) suggest using attention maps. These maps compress the channel dimension of the feature maps, using an attention operator $\mathcal{A}$, which maps from $\mathbb{R}^{c \times h \times w}$ to $\mathbb{R}^{h \times w}$. They propose the following two variants

$$\mathcal{A}_{\text{sum}}^p(x, \theta, l) = \sum_{i=1}^{c} |f_{l,i}(x, \theta)|^p, \ \ \mathcal{A}_{\text{mean}}^p(x, \theta, l) = \frac{1}{c} \sum_{i=1}^{c} |f_{l,i}(x, \theta)|^p \tag{24}$$

where $f_{l,i}$ is the $i^{\text{th}}$ channel activation of $x$ at the $l^{\text{th}}$ layer and $p > 1$. To perform KD within the teacher-student framework, Li et al. (2021a) first fine-tune $\theta$ using $\mathcal{D}_m$ resulting in a teacher model with parameters $\theta_T$. The student's parameters, $\theta_S$, are the original parameters $\theta$ that have not been fine-tuned. The KD is then performed by comparing the activation maps between the teacher and student models. To achieve this, Li et al. (2021a) use $\mathcal{A}_{sum}^p$ to define a distillation loss as

$$\mathcal{L}_{\text{NAD}}(x, \theta_T, \theta_S, l) = \left\| \frac{\mathcal{A}(x, \theta_T, l)}{\|\mathcal{A}(x, \theta_T, l)\|_2} - \frac{\mathcal{A}(x, \theta_S, l)}{\|\mathcal{A}(x, \theta_S, l)\|_2} \right\|_2, \tag{25}$$

and solve the following optimization problem

$$\min_{\theta_S} \mathcal{L}_{\text{CE}}(\mathcal{D}_m, \theta_S) + \frac{\lambda}{|\mathcal{D}_m|} \sum_{(x,y) \in \mathcal{D}_m} \sum_{l=1}^{L} \mathcal{L}_{\text{NAD}}(x, \theta_T, \theta_S, l), \tag{26}$$

where $\lambda$ controls the contribution of the distillation loss. According to Li et al. (2021a), the inclusion of $\mathcal{L}_{\text{NAD}}$ helps regularize $\theta$ by aligning the activation maps of $\theta_S$ and $\theta_T$ thus removing the backdoor behavior. However, since knowledge is distilled from a fine-tuned version of $\theta$, the effectiveness of this approach is unclear if fine-tuning does not successfully remove the backdoor. This concern was underscored by the initial findings of both FST and FT-SAM, where fine-tuning alone proved ineffective at eliminating the backdoor.

### 4.2.2 BCU

In contrast to NAD Li et al. (2021a), Pang et al. (2023) propose using softmax probabilities $a$ of $x$ to perform KD rather than attention maps. Specifically, Pang et al. (2023) compare the temperature-scaled softmax probability scores $\tilde{a}_T = f(x, \theta_T)$ and $\tilde{a}_S = f(x, \theta_S)$, produced by $\theta_S$ and $\theta_T$, respectively, to facilitate KD. To compare $\tilde{a}_T$ and $\tilde{a}_S$, Pang et al. (2023) employ KL-Divergence $\mathcal{L}_{\text{KL}}$ and solve the following optimization problem

$$\min_{\theta_S} \mathcal{L}_{\text{KL}}(\tilde{a}_T, \tilde{a}_S). \tag{27}$$

Rather than fine-tuning $\theta$ to produce $\theta_T$, Pang et al. (2023) set $\theta_T = \theta$ and reinitialize a subset of the student parameters $\theta_S$. Unlike FST Min et al. (2024), Pang et al. (2023) uniformly reinitialize a proportion $n$ ($0 \le n \le 1$) of the parameters within each layer, with $n$ increasing for deeper layers. They identify this reinitialization strategy as the best approach through a series of experiments. The intended effect is to impair the student's ability to perform both clean and backdoor tasks before distillation, so that the subsequent optimization transfers clean-task knowledge from the teacher while weakening the link between the trigger and the backdoor task.

Unlike previous proposals, the use of KL-divergence by Pang et al. (2023) makes their approach dataset-agnostic. As a result, a defender can use any labeled or unlabeled in- or out-of-distribution dataset compatible with their model (i.e., having the same input dimensionality) to perform backdoor mitigation.

### 4.3 Synthesis Unlearn

The fine-tuning approaches discussed thus far use $\mathcal{D}_m$ to fine-tune $\theta$. An alternative strategy involves synthesizing a set of surrogate backdoor data $(\tilde{x}, \tilde{y}) \in \widetilde{\mathcal{D}}_m$, which is used alongside $\mathcal{D}_m$ for fine-tuning. Here, $\tilde{x}$ and $\tilde{y}$ represent the surrogate backdoor data. The inclusion of this initial synthesis step allows these approaches to exploit information from $\widetilde{\mathcal{D}}_m$ to directly unlearn the backdoor task.

#### 4.3.1 MESA

To synthesize $\tilde{x}$, Qiao et al. (2019) propose training a generative model $G$, parameterized by $\gamma$, to replicate the trigger distribution used by the adversary. To train $G$, they introduce a new maximum-entropy staircase approximation algorithm. This algorithm trains $G$ as a combination of $n$ sub-models that collectively generate a candidate trigger for a given input $x$. However, using a set of sub-models to train $G$ requires the defender to know the trigger's position, approximate size, and the backdoor target. The optimization problem they solve is

$$\min_\theta \frac{1}{|\mathcal{D}_m|} \sum_{x,y \in \mathcal{D}_m} [\lambda \mathcal{L}_{\mathrm{CE}}(x, y, \theta) + (1 - \lambda)\mathcal{L}_{\mathrm{CE}}(\tilde{x}, y, \theta)] \tag{28}$$

where $\tilde{x} = x + G(\gamma)$ and $\lambda \in [0, 1]$ is a hyperparameter selected by the defender to control how many elements of $\mathcal{D}_m$ have $\delta$ applied. This approach aims to balance restoring the classification of $\tilde{x}$ to $y$ while preserving the original classification performance.

#### 4.3.2 BAERASER

Inspired by MESA Qiao et al. (2019), Liu et al. (2022) adopt the same synthesis method for generating surrogate backdoor data but proposes a different fine-tuning step. Using $G$, they generate a surrogate backdoor dataset $\widetilde{\mathcal{D}}_m$ that is used in conjunction with $\mathcal{D}_m$ to unlearn the backdoor task. The surrogate dataset is defined as $\widetilde{\mathcal{D}}_m = \{(\tilde{x}, \tilde{y}) \mid \tilde{x} = x + \delta \mid \delta = G(x, \gamma) \mid (x, y) \in \mathcal{D}_m\}$, assuming the defender has access to $\tilde{y}$. To perform unlearning, Liu et al. (2022) solve the following optimization problem

$$\min_\theta \lambda_1 [\mathcal{L}_{\mathrm{CE}}(\mathcal{D}_m, \theta) - \mathcal{L}_{\mathrm{CE}}(\widetilde{\mathcal{D}}_m, \theta)] + \lambda_2 \sum_{l=1}^{L} w_l \|\theta_l - \bar{\theta}_l\|_1, \tag{29}$$

where $\lambda_1$ and $\lambda_2$ control the strength of the two loss terms. The first term encourages misclassification of $\widetilde{\mathcal{D}}_m$ by subtracting its loss from that of $\mathcal{D}_m$. However, this term is unbounded and can dominate the optimization after a few iterations. The second loss term regularizes the solution by minimizing the layer-wise distance between $\theta$ and the original value $\bar{\theta}$, using a layer-wise scalar weight $w_l$.

#### 4.3.3 NC

Unlike both MESA and BAEARSER, Wang et al. (2019a) propose a method that removes the assumption of knowing the backdoor target and the approximate size and position of $\rho$. To achieve this, Wang et al. (2019a) learn an input perturbation $\delta$ that replaces specific image pixels using a binary mask $\mathbf{m}$. Here, $\delta$ and $\mathbf{m}$ are 3D and 2D matrices, respectively, with width and height dimensions matching $x$. To apply $\delta$ to $x$ given $\mathbf{m}$, the function $A(x, \mathbf{m}, \delta) \to \hat{x}$ is used where if $\mathbf{m}_{j,i} = 1$, $A$ replaces the pixel in the $j^{\mathrm{th}}$ row and $i^{\mathrm{th}}$ column of $x$ with the corresponding value in $\delta$. To learn $\delta$ and $\mathbf{m}$, Wang et al. (2019a) solve the following optimization problem

$$\min_{\mathbf{m}, \delta} \sum_{x \in \mathcal{D}_m} \mathcal{L}_{\mathrm{CE}}(A(x, \mathbf{m}, \delta), t) + \lambda \|\mathbf{m}\|_1, \tag{30}$$

where $\lambda$ controls the strength of the second regularization term, which promotes sparsity in the solution for $\mathbf{m}$. Since $t$ is unknown to the defender, a unique solution for $\delta$ and $\mathbf{m}$ is determined separately for

each class. An anomaly detection mechanism, using median absolute deviation, is then employed to identify anomalous class pairs. If such a pair is found, the proposed approach prunes the final dense layer of the model to mitigate the effect of the backdoor. To perform model pruning, the TAC metric [cf. equation 3] is used. Neurons that exhibit the largest average activation difference when $\delta$ is applied to $D_m$ using $A$ are iteratively pruned. Despite relying on model pruning, the method shares key similarities with MESA and BAEARSER, making it relevant to this section.

### 4.3.4 BTI-DBF

Rather than view pruning and fine-tuning as two separate approaches, Xu et al. (2024) considers how they can be used in combination. More specifically, they use a three-stage process to first decouple the benign and backdoor features through feature masking, then learn a generator network to synthesize triggers, and finally fine-tune the original model parameters using the generator network.

During the first stage, a soft mask $\mathbf{m}$ is learned and applied to the feature map produced by the final feature extraction layer. To formalize this, let $f(x, \theta; \mathbf{m})$ denote the model output when the feature map is element-wise multiplied by $\mathbf{m}$. To learn $\mathbf{m}$ given clean data $\mathcal{D}_m$, they solve the following optimization problem:

$$\min_{\mathbf{m}} \mathcal{L}_{\text{CE}}(\mathcal{D}_m, \theta; \mathbf{m}) - \mathcal{L}_{\text{CE}}(\mathcal{D}_m, \theta; (1 - \mathbf{m})). \tag{31}$$

The objective encourages the features selected by $\mathbf{m}$ to be sufficient for correct classification, while encouraging the remaining features (selected by $1 - \mathbf{m}$) to be insufficient.

Using the learned mask $\mathbf{m}$, a generator network $G : \mathcal{X} \rightarrow \mathcal{X}$ is then trained to generate perturbed versions of $D_m$, denoted as $\widetilde{\mathcal{D}}_m = \{(\tilde{x}, y) \mid \tilde{x} = G(x) \text{ for } (x, y) \in \mathcal{D}_m\}$. To train $G$, they solve the following optimization problem:

$$
\begin{aligned}
\min_{G} \quad & \|(f_l(x, \theta) - f_l(G(x), \theta)) \odot \mathbf{m}\|_2 - \|(f_l(x, \theta) - f_l(G(x), \theta)) \odot (1 - \mathbf{m})\|_2, \\
s.t. \quad & \|x - G(x)\|_2 \leq \tau, \ \forall (x, y) \in \mathcal{D}_m,
\end{aligned} \tag{32}
$$

where $f_l$ is the feature map from the last feature extraction layer. This objective trains the generator to create a poisoned sample $G(x)$ whose representation is similar to the original sample $x$ in the benign features (masked by $\mathbf{m}$) but dissimilar in the presumed backdoor features (masked by $1 - \mathbf{m}$).

Finally, using $G$, the original set of model parameters $\theta$ is fine-tuned by solving the following optimization problem:

$$\min_{\theta} \mathcal{L}_{\text{CE}}(\mathcal{D}_m, \theta) + \mathcal{L}_{\text{CE}}(\widetilde{\mathcal{D}}_m, \theta) + \sum_{(x,y) \in \mathcal{D}_m} \|f_l(x, \theta) - f_l(G(x), \theta)\|_2. \tag{33}$$

Together, Xu et al. (2024) claim that these stages decouple benign and backdoor features, use this decoupling to learn a trigger generator, and then leverage the generator to unlearn the backdoor associations. While this approach is unique, we highlight a few critical limitations. First, the decoupling and generative training steps each use a negative term that acts as a maximization objective. Although both terms are indirectly bounded by constraints (i.e., $\mathbf{m} \in [0, 1]$ and $\|x - G(x)\|_2 \leq \tau$), the dominance of each term within Equations 31 and 32 across each iteration is not evaluated. Moreover, in Equation 32, classification differences are not strictly enforced, as the $\ell_2$ norm between the channels selected by $1 - \mathbf{m}$ is maximized.

### 4.4 Adversarial Training

Instead of approximating $\delta$ in a discrete step, recent works have incorporated concepts from adversarial training to perform mitigation. In essence, these approaches alternate between an adversarial objective and a mitigation objective. However, the critical distinction lies in the design of the adversarial objective.

Unlike traditional adversarial examples, the adversarial objective is specifically tailored to generate surrogate backdoor images. This ensures that the outer mitigation objective remains effective, allowing the model to unlearn the backdoor task while preserving performance on the clean task.

### 4.4.1 PBE

In Mu et al. (2023), the authors explore the behavior of untargeted adversarial attacks on backdoored models. Given $x$ and $y$, they generate an input perturbation $\delta$ by solving the following adversarial optimization problem

$$\max_{\|\delta\|_2 \leq \epsilon} \mathcal{L}_{\text{CE}}(\tilde{x}, y, \theta), \tag{34}$$

where $\epsilon$ controls the strength of the perturbation and $\tilde{x} = x + \delta$. Upon analyzing the classification of $\tilde{x}$, Mu et al. (2023) observed that a backdoored model tends to classify $\tilde{x}$ as the backdoor target, whereas a benign model produces a uniform distribution (see (Mu et al., 2023, Fig. 4)). Hence, they hypothesized that $\tilde{x}$ interacts with the backdoored model similarly to $\hat{x}$, the actual backdoor version of $x$. However, it is important to note that the proportion of samples classified as the target class in (Mu et al., 2023, Fig. 4) does not exceed 61%.

To exploit this observation, Mu et al. (2023) propose a fine-tuning strategy where $\theta$ is trained using both $\mathcal{D}_m$ and $\widetilde{\mathcal{D}}_m = \{(\tilde{x}, y) \mid \tilde{x} = x + \delta \mid (x, y) \in \mathcal{D}_m\}$. They alternate between solving the following two optimization problems

$$\min_{\theta} \mathcal{L}_{\text{CE}}(\mathcal{D}_c, \theta), \quad \min_{\theta} \mathcal{L}_{\text{CE}}(\widetilde{\mathcal{D}}_m, \theta), \tag{35}$$

where $\delta$ is computed using the PGD attack Madry et al. (2018).

### 4.4.2 i-BAU

In Zeng et al. (2022), the authors propose a redesigned adversarial objective aimed at identifying a universal input perturbation $\delta$, which functions similarly to AWM Chai & Chen (2022). Here, universal refers to a perturbation that applies to all elements within $\mathcal{D}_m$. Using $\widetilde{\mathcal{D}}_m = \{(\tilde{x}, y) \mid \tilde{x} = x + \delta \mid (x, y) \in \mathcal{D}_m\}$, Zeng et al. (2022) set up the following minimax optimization problem

$$\min_{\theta} \max_{\|\delta\|_2 \leq \epsilon} \mathcal{L}_{\text{CE}}(\widetilde{\mathcal{D}}_m, \theta). \tag{36}$$

However, their experiments (see (Zeng et al., 2022, Fig. 1)) reveal that solving this minimax problem directly often yields unstable and unreliable results. This instability is attributed to the inner maximization step failing to find an optimal solution for $\delta$. To alleviate this issue, Zeng et al. (2022) propose solving the outer minimization step using the following gradient

$$\nabla_{\theta} \mathcal{L}_{\text{CE}}(\widetilde{\mathcal{D}}_m, \theta) + (\nabla \delta)^T \nabla_{\delta} \mathcal{L}_{\text{CE}}(\widetilde{\mathcal{D}}_m, \theta), \tag{37}$$

where $\nabla \delta$ is the *response Jacobian* of the inner maximization problem. Given that the inner maximization step produces a suboptimal solution for $\delta$, they calculate $\nabla \delta$ as

$$\nabla \delta = - \left( \nabla_{\delta}^2 \mathcal{L}_{\text{CE}}(\widetilde{\mathcal{D}}_m, \theta) \right)^{-1} \nabla_{\delta, \theta}^2 \mathcal{L}_{\text{CE}}(\widetilde{\mathcal{D}}_m, \theta). \tag{38}$$

This ensures that the response Jacobian captures the sensitivity of $\delta$ to changes in $\theta$, while $\nabla_{\delta} \mathcal{L}_{\text{CE}}(\widetilde{\mathcal{D}}_m, \theta)$ captures the direct sensitivity of $\delta$. These adjustments allow the gradient update for $\theta$ to incorporate the sensitivity of $\delta$, resulting in a more stable and reliable solution for adversarial fine-tuning. The complexity of the gradient estimation method in Zeng et al. (2022) increases the likelihood of overfitting, as $\nabla_{\delta}$ is dependent

on the estimation of $\nabla \delta$. Notably, $\nabla \delta$ requires estimation using second-order algorithms. While Zeng et al. (2022) asserts that these methods are robust to inaccuracies in the Hessian, the referenced literature assumes access to a large training dataset.

### 4.4.3 SAU

To enhance existing approaches, Wei et al. (2023) propose filtering candidate perturbations $\delta$ based on their ability to induce consistent misclassification across two classifiers. In this context, consistent misclassification means that both classifiers classify $\tilde{x}$ as the same incorrect class $\tilde{y}$. Formally, Wei et al. (2023) optimize $\delta$ for each $(x, y) \in \mathcal{D}_m$ by solving the following optimization problem

$$\max_{\|\delta\|_p \leq \epsilon} \left\{ \frac{\lambda_1}{2} [\mathcal{L}_{\mathrm{CE}}(\tilde{x}, y, \theta) + \mathcal{L}_{\mathrm{CE}}(\tilde{x}, y, \bar{\theta})] - \lambda_2 \, \mathrm{JS}(f(\tilde{x}, \theta), f(\tilde{x}, \bar{\theta})) \right\}, \tag{39}$$

where JS is the Jensen-Shannon divergence, and $\lambda_1$ and $\lambda_2$ are hyperparameters. Because the defender does not have access to two independently trained classifiers, SAU uses the original model parameters $\bar{\theta}$ as the second classifier and keeps $\bar{\theta}$ fixed. Unlike PBE and NPD, SAU aims to distinguish between adversarial examples and backdoor triggers. To achieve this, they ensure that $\delta$ causes both $\theta$ and $\bar{\theta}$ to misclassify $\tilde{x}$ consistently, a behavior more characteristic of backdoor triggers than typical adversarial examples (see (Wei et al., 2023, Fig. 2)).

Once the surrogate backdoor dataset $\widetilde{\mathcal{D}}_m = \{(\tilde{x}, x, \tilde{y}, y) \mid \tilde{y} \leftarrow f(\tilde{x}, \theta) \mid \tilde{x} = x + \delta \mid (x, y) \in \mathcal{D}_m\}$ is generated, Wei et al. (2023) solve the following optimization problem to fine-tune the model

$$\min_{\theta} \left\{ \frac{1}{|\widetilde{\mathcal{D}}_m|} \sum_{(\tilde{x}, x, \tilde{y}, y) \in \widetilde{\mathcal{D}}_m} \lambda_3 \, \mathcal{L}_{\mathrm{CE}}(x, y, \theta) - I(\tilde{y} \neq y) \log[1 - s_{\tilde{y}}(f(\tilde{x}, \theta))] \right\}, \tag{40}$$

where $I$ is the indicator function that is 1 if $\tilde{x}$ is misclassified and $\lambda_3$ is another hyperparameter. This formulation balances the performance on $\mathcal{D}_m$, represented by the first term, with correcting the classification of $\tilde{x}$ to $y$, captured by the second term.

### 4.5 Summary

Fine-tuning, as an alternative to model pruning for backdoor mitigation, adjusts the parameters of a model rather than removing them. This strategy is typically governed by an objective function incorporating tailored regularization terms. Conventional fine-tuning approaches, such as FST and FT-SAM, attempt to adjust model weights to eliminate backdoors, though they often struggle with escaping local minima and thus fail to fully mitigate the backdoor threat. More advanced approaches based on KD, such as NAD and BCU, reframe fine-tuning as a process of filtering out harmful information. These approaches leverage the distillation of knowledge from a teacher model to a student model to effectively mitigate backdoor attacks while retaining the model's performance on clean data. Additionally, approaches such as MESA and BAERASER employ surrogate data generation to support the fine-tuning process. However, these approaches rely on the defender having prior knowledge about the specific trigger used by an adversary. To remove this assumption, approaches such as PBE, i-BAU, and SAU modify existing adversarial training techniques. Similar to pruning approaches, each fine-tuning approach presents unique strengths and weaknesses, necessitating thorough evaluation across a diverse range of settings to fully understand their effectiveness.

## 5 Experimental Setup

In this section, we describe the experimental setup for evaluating 19 of the 23 approaches discussed in Sections 3 and 4. We benchmark each approach across a wide range of settings spanning backdoor attacks, model architectures, datasets, and poisoning ratios, resulting in 288 distinct attack scenarios. Unlike Wu et al. (2022), we also test each considered mitigation approach across three data-availability settings, leading to over 120,000 individual experiments in total. We use the *BackdoorBench* toolkit Wu et al. (2022), which

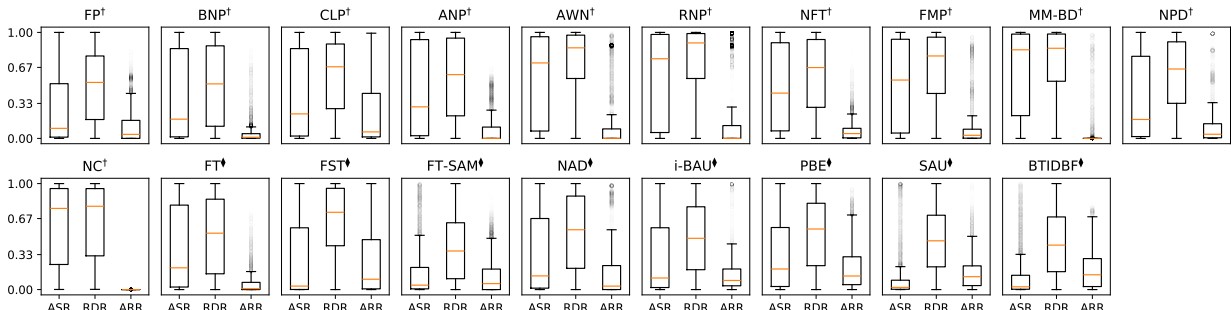

Figure 3: Box plots of the ASR, RDR, and ARR results for each approach across all considered settings. †
= Pruning and ♦ = Fine-tuning. Note: NC results are only for CIFAR-10.

provides most of the required functionality, and extend it with implementations of five additional mitigation approaches.

### 5.1 Attacks

Our evaluations include all backdoor attacks introduced in Section 2.2, except LIRA. We exclude LIRA because it produced weak attacks in preliminary experiments, which would make mitigation results difficult to interpret. The evaluated attacks are therefore BadNets Gu et al. (2017), Blended Chen et al. (2017), Signal Barni et al. (2019), LF Zeng et al. (2021), SSBA Li et al. (2021b), IAB Nguyen & Tran (2020), BPP Wang et al. (2022), and WaNet Nguyen & Tran (2021). For each attack, we use the default configurations provided in *BackdoorBench*. We evaluate poisoning ratios of 1%, 5%, and 10%, selected based on the settings reported in Wu et al. (2022).

### 5.2 Mitigation Methods

We evaluate all approaches discussed in Sections 3 and 4 except MESA, BAERASER, BCU, and UNIT. The benchmarked methods are FP Liu et al. (2018), BNP Zheng et al. (2022a), CLP Zheng et al. (2022b), ANP Wu & Wang (2021), AWM Chai & Chen (2022), RNP Li et al. (2023), NFT Karim et al. (2024), FMP Huang & Bu (2023), MM-BD Wang et al. (2023), NPD Zhu et al. (2024b), FT Liu et al. (2018), FST Min et al. (2024), FT-SAM Zhu et al. (2023), NAD Li et al. (2021a), NC Wang et al. (2019a), BTI-DBF Xu et al. (2024), PBE Mu et al. (2023), i-BAU Zeng et al. (2022), and SAU Wei et al. (2023). We evaluate NC only on CIFAR-10 because its computational cost scales with the number of classes. We exclude MESA Qiao et al. (2019) and BAERASER Liu et al. (2022) because they require additional assumptions. We exclude BCU because it assumes access to an out-of-distribution dataset; a fair evaluation would require additional dataset choices, making the benchmark computationally prohibitive. Finally, because MM-BD and UNIT are methodologically similar, we evaluate MM-BD as the representative method from this pair.

For approaches already implemented in BackdoorBench, we use the default configurations. We implement AWM, MM-BD, RNP, NFT, FMP, BTI-DBF, FST and PBE using the code provided by the authors and the training configurations reported in each paper. In cases where hyperparameter values are not explicitly reported, we use the values from the respective codebase. In the Supplementary Materials (Table I), we summarize the hyperparameter values used for each approach.

### 5.3 Other Settings

We consider three datasets: CIFAR-10, German Traffic Sign Recognition Benchmark (GTSRB), and Tiny-ImageNet, containing 10, 43, and 200 classes, respectively. To evaluate the effect of data availability on the performance of each approach, we assess them under three data settings based on the samples-per-class (SPC) value. Specifically, we evaluate each approach considering SPC values of 2, 10, and 100. For each data setting, we conduct 10 iterations, with each iteration using a different random data partition. Moreover,

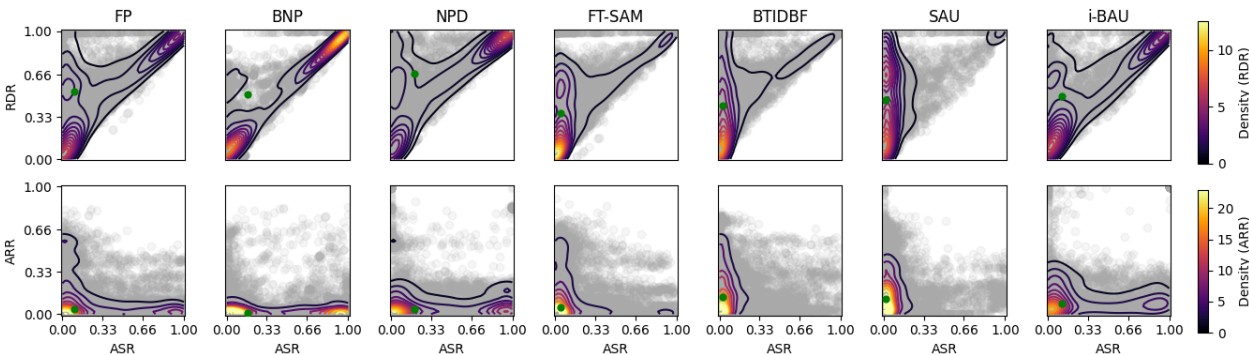

Figure 4: Scatter plots of the ASR, RDR, and ARR results for the best performing approach across all considered settings.

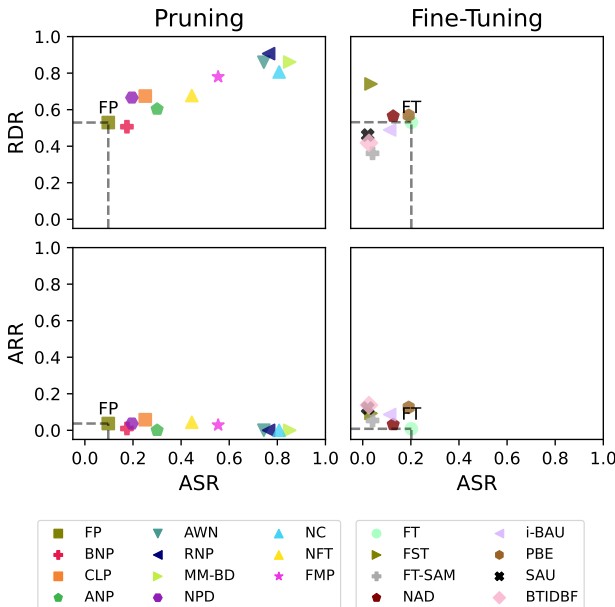

Figure 5: Scatter plot of the median RDR and ARR versus ASR for each approach across all considered settings. Note: NC results are only for CIFAR-10.

we employ four model architectures: PreAct-ResNet18 (ResNet), VGG-19 with batch normalization (VGG), EfficientNet-B3 (EfficientNet), and MobileNetV3-Large (MobileNet), using their default configurations as provided by *BackdoorBench*.

### 5.4 Performance Measures

To evaluate mitigation effectiveness, we use three standard performance measures: clean accuracy (ACC), attack success rate (ASR), and recovery accuracy (RA). Although these metrics are sometimes named differently across papers, they capture the following quantities:

**ACC:** accuracy on clean test data, measuring preservation of the original classification task.

**ASR:** accuracy on triggered test data when labels are changed to the attack target, measuring how often the attack succeeds. Samples whose original label is already the target class are omitted.

**RA:** accuracy on triggered test data using the original labels, measuring whether the model correctly classifies triggered samples after mitigation.

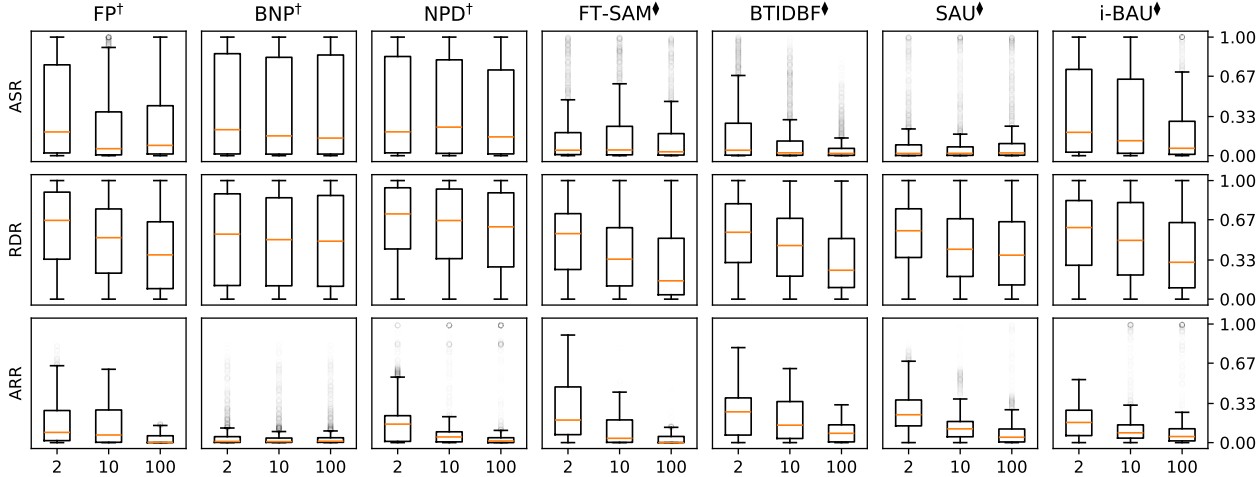

Figure 6: Box plots of the ASR, RDR, and ARR results for each approach and SPC values of 2, 10, and 100. † = Pruning and ♦ = Fine-tuning.

ASR therefore measures targeted misclassification, whereas RA measures recovery of the original task behavior on triggered inputs. In our evaluation, ASR is reported directly, while ACC and RA are converted to normalized difference ratios so that pre- and post-mitigation behavior can be compared across attacks, datasets, and architectures.

### 5.4.1 Accuracy Reduction Ratio (ARR)

To quantify the impact of mitigation on the accuracy of the original classification task, we take into account the accuracy values before and after mitigation, denoted as $\text{ACC}_{\text{pre}}$ and $\text{ACC}_{\text{post}}$, respectively. Therefore, we calculate the accuracy reduction ratio (ARR) as

$$\text{ARR} = \frac{\text{ACC}_{\text{pre}} - \text{ACC}_{\text{post}}}{\text{ACC}_{\text{pre}}}. \tag{41}$$

Dividing the difference by the pre-mitigation accuracy accounts for variations in $\text{ACC}_{\text{pre}}$. For instance, most Tiny-ImageNet models exhibit lower $\text{ACC}_{\text{pre}}$ compared to their CIFAR-10 counterparts. Ideally, this ratio approaches 0, indicating minimal reduction in accuracy due to mitigation.

### 5.4.2 Recovery Difference Ratio (RDR)

To measure the effectiveness of a mitigation strategy in restoring the classification of backdoor samples to their original classes, we calculate the recovery difference ratio (RDR) as

$$\text{RDR} = \frac{\text{ACC}_{\text{pre}} - \text{RA}_{\text{post}}}{\text{ACC}_{\text{pre}}}. \tag{42}$$

Like ARR, RDR normalizes by the pre-mitigation clean accuracy. An ideal mitigation method restores the recovery accuracy on triggered inputs to the pre-mitigation clean accuracy, so $\text{RA}_{\text{post}} \approx \text{ACC}_{\text{pre}}$ and RDR approaches zero. As with ASR and ARR, lower values indicate better performance.

## 6 Evaluation Results

In this section, we present the results of our evaluation across the considered settings. We first discuss the overall performance of the mitigation approaches, then analyze the effect of data availability, attack type, model architecture, and dataset.

### 6.1 Overall Results

Figures 3 and 4 summarize the results for each considered approach. Figure 3 uses box plots to show the distributions of ASR, RDR, and ARR, while Figure 4 plots each method's ARR and RDR against ASR. In Figure 3, the top row contains the pruning approaches and NC, while the second row contains the remaining fine-tuning approaches. Figure 5 plots each method's median ARR and RDR against its median ASR. We draw rectangles using the median ARR, RDR, and ASR values of FP and FT; a method must fall within the relevant rectangle to improve on the corresponding baseline across all three metrics. The optimal point is the bottom-left corner in all panels. FP and FT serve as the baseline pruning and fine-tuning approaches, respectively.

### 6.1.1 Pruning Methods

Figure 5 shows that none of the evaluated pruning approaches lies inside the bottom-left rectangle defined by FP's median ARR, RDR, and ASR values. Thus, no evaluated pruning method improves on FP across all three median metrics. Nevertheless, ANP, BNP, CLP, and NPD perform comparably to FP across the three measures.

Metric-based pruning approaches (FP, BNP, CLP, and FMP) show limited overall effectiveness. Although they achieve low median ASR values, Figure 4 shows heavy ASR tails, indicating failures in a non-negligible subset of settings. For ARR, FP, FMP, and BNP perform well, with low medians and small variances. CLP has a median ARR comparable to FP, BNP, and FMP but substantially higher variance. By contrast, NC and FMP perform worse than FP on ASR and RDR, even when their clean-accuracy cost is relatively modest.

Masking-based pruning approaches (i.e., ANP, AWM, RNP and NFT) also demonstrate limited overall effectiveness. ANP's distribution of ASR, RDR, and ARR values is similar to FP, but with increased variation in ASR and RDR. While RNP and AWM perform well in terms of ARR, their high median ASR and RDR offset this benefit. When comparing ANP with AWM and RNP, we find that, although AWM and RNP have been designed to improve upon ANP, in our evaluations, ANP consistently outperforms both. Finally, NFT falls between ANP and AWM/RNP in terms of ASR, while exhibiting comparable RDR median and variance to ANP.

Additive pruning methods (i.e., MM-BD and NPD) also show limited effectiveness. Similar to RNP, MM-BD has high median ASR and RDR values. Although NPD appears to outperform MM-BD overall, it fails to improve upon FP in any performance measure.

### 6.1.2 Fine-Tuning Methods

In contrast to model-pruning approaches, FT-SAM, SAU, i-BAU, and BTI-DBF improve on FT in terms of median ASR, and several also improve on median RDR. However, none improves on FT across ASR, RDR, and ARR simultaneously because the ASR/RDR gains are accompanied by ARR trade-offs. FST, NAD, and PBE do not surpass FT overall, although NAD and, to some extent, PBE perform comparably to FT.

Among the conventional fine-tuning approaches (i.e., FST and FT-SAM), only FT-SAM outperforms FT, as indicated by its lower median ASR and RDR, as well as reduced variance, in Figure 3. However, this improvement comes at the expense of a higher median ARR and increased variance. While FST achieves improved ASR relative to FT, it underperforms in terms of RDR. Additionally, FST exhibits higher variance in ASR and ARR compared to FT. NAD demonstrates overall performance comparable to FT but with increased variance in ARR.

Adversarial training approaches (i.e., PBE, i-BAU, and SAU) exhibit varied performance. Compared to FT, i-BAU shows lower median ASR and RDR, with slightly higher ARR median and variance. PBE achieves similar ASR and RDR compared to FT, though it exhibits a higher median ARR and a longer tail. In contrast, SAU demonstrates significant improvement over FT, with a lower median ASR and reduced variance. However, similar to FT-SAM, this improvement comes with a trade-off in ARR performance.

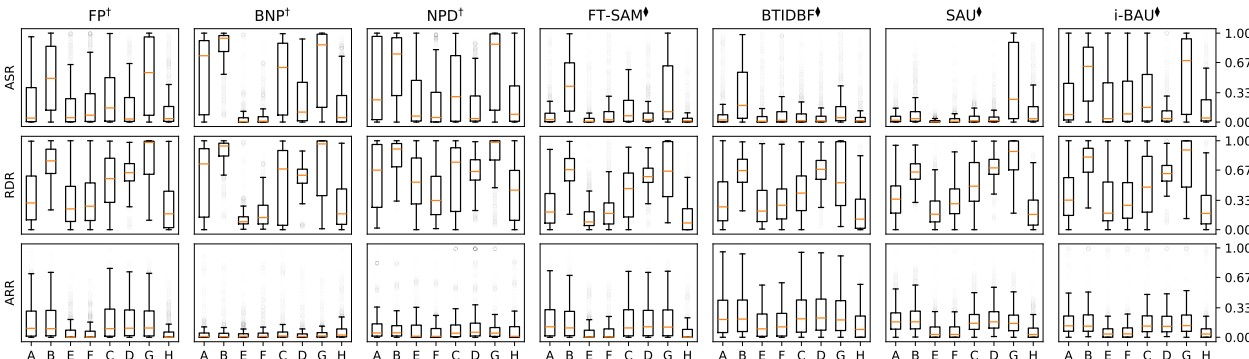

Figure 7: Box plots of the ASR, RDR, and ARR results for the selected approaches and different attack types. † = Pruning and ♦ = Fine-tuning. A = BadNet, B = Blended, C = LF, D = Signal, E = BPP, F = IAB, G = SSBA and H = WaNet.

Finally, BTI-DBF exhibits similar performance to SAU, with a low median ASR and significantly reduced variance compared to FT. Moreover, its RDR median and variance are comparable to FT-SAM and SAU.

### 6.1.3 Density Analysis

The density results in Figure 4 reveal two important patterns when comparing pruning-based methods (FP, BNP, and NPD) with fine-tuning approaches (FT-SAM, BTI-DBF, SAU, and i-BAU). First, pruning methods generally trade off RDR and ASR in a near 1:1 manner. For example, comparing BNP and SAU highlights that BNP's RDR performance is closely coupled with ASR, whereas SAU exhibits lower ASR variance but substantially higher RDR variance. Second, this distinction is critical: a defense that achieves low ASR but high or unstable RDR does not meaningfully restore correct classification. In such cases, images may no longer be misclassified into the attacker's target class, but they remain misclassified into other incorrect classes. Thus, reducing ASR without simultaneously stabilizing RDR offers limited practical benefit.

### 6.1.4 Summary and Comparison with Baselines

A central claim of this survey is that, despite a steady stream of new mitigation methods, improvements over the seminal 2018 baselines FP and FT remain modest across the full range of evaluation settings. Figure 5, which plots each method's median ARR and RDR against its median ASR, demonstrates this. The rectangle defined by FP's median values bounds the region in which a pruning method would improve on all three metrics simultaneously, and no evaluated pruning method enters this rectangle. The analogous rectangle for FT is entered by SAU, FT-SAM, i-BAU, and BTI-DBF on ASR, but none of these methods enters all three metrics simultaneously.

The box plots in Figure 3 reinforce this view from a different angle. Except for SAU, FT-SAM, i-BAU, and BTI-DBF, the evaluated approaches exhibit substantial variability across the full range of tested settings, especially in ASR and RDR. SAU, FT-SAM, i-BAU, and BTI-DBF achieve lower median ASR with reduced variance, but their gains are accompanied by ARR trade-offs and persistently weak or unstable RDR. The overall picture is therefore one of localized progress: recent methods improve particular metrics or settings, but none consistently improves across the full evaluation space. This matters in practice because a triggered image that is no longer mapped to the attack target can still be misclassified into another incorrect class. Future research should therefore treat RDR as a central metric and report variability, not only median performance.

In the following subsections, we discuss BNP and NPD as they are the best-performing pruning approaches. Similarly, we discuss FT-SAM, BTI-DBF, SAU, and i-BAU as the top-performing fine-tuning approaches.

## 6.2 Data Availability

To assess the effect of data availability, we evaluate each approach using 2, 10, and 100 samples per class (SPC). Figure 6 summarizes the ASR, RDR, and ARR distributions for the strongest methods identified above. Overall, reducing SPC worsens the median values of most methods, although pruning-based approaches tend to be more stable than fine-tuning-based approaches under limited mitigation data.

### 6.2.1 Model-Pruning

Data availability has a smaller impact on most pruning methods than on fine-tuning methods. BNP is particularly stable across SPC settings. FP and NPD are more affected by reduced SPC, mainly through a noticeable increase in median ARR, while their ASR and RDR changes are comparatively minor.

### 6.2.2 Fine-Tuning

Unlike the evaluated pruning approaches, SPC has a greater impact on the performance of fine-tuning approaches. Specifically, median ARR and RDR increase significantly for FT-SAM and BTI-DBF when SPC is reduced. Although a similar trend is observed for SAU, the impact of SPC is less pronounced.

## 6.3 Backdoor Attack

Figure 7 presents the results for the selected approaches across the eight considered attacks. Most approaches vary substantially across attack types, especially in ASR and RDR. For example, BNP performs well against BPP but poorly against Blended. FT-SAM, BTI-DBF, and SAU are more consistent in ASR, although important exceptions remain: SSBA for SAU, Blended for BTI-DBF, and Blended and SSBA for FT-SAM. Moreover, the RDR performance of FT-SAM and SAU still fluctuates across attacks, reinforcing the distinction between suppressing the target-class attack and restoring correct classification.

Among the tested attacks, Blended and SSBA are the most difficult to mitigate, as reflected by higher median ASR values and greater variance across most approaches. BadNets also produces substantial ASR and RDR variation for most methods, except FT-SAM and SAU, which is notable given its role as the foundational backdoor attack. Finally, although recent methods such as NPD and SAU explicitly target dynamic backdoor behavior, performance on SSBA remains weak, showing that dynamic-trigger robustness is still unresolved.

## 6.4 Model Architecture

Figure 8 shows the results for each selected approach across the four tested model architectures. Except for SAU, most approaches exhibit inconsistent performance across the considered architecture types, as indicated by fluctuations in ASR or ARR values. Notably, SAU demonstrates the most consistent performance across all architectures.

For FP, ARR variance increases significantly when using the EfficientNet architecture. Similarly, although BNP's ARR performance remains mostly stable, its median RDR and ASR show considerable variation. For NPD, median ASR noticeably increases when the MobileNet architecture is used.

While FT-SAM, BTI-DBF, and i-BAU maintain relatively consistent median ARR and ASR across architectures, the distribution tails expand in some cases. For FT-SAM, the ARR and ASR tails appear to trade off: settings with a smaller ARR tail often show a longer or heavier ASR tail, and vice versa. In contrast, SAU exhibits only minor architectural variation across all three measures.

## 6.5 Dataset

Figure 9 shows the results for each selected approach across the three considered datasets. Similar to the performance variability observed with different model architectures, there is noticeable variability across datasets, often more pronounced.

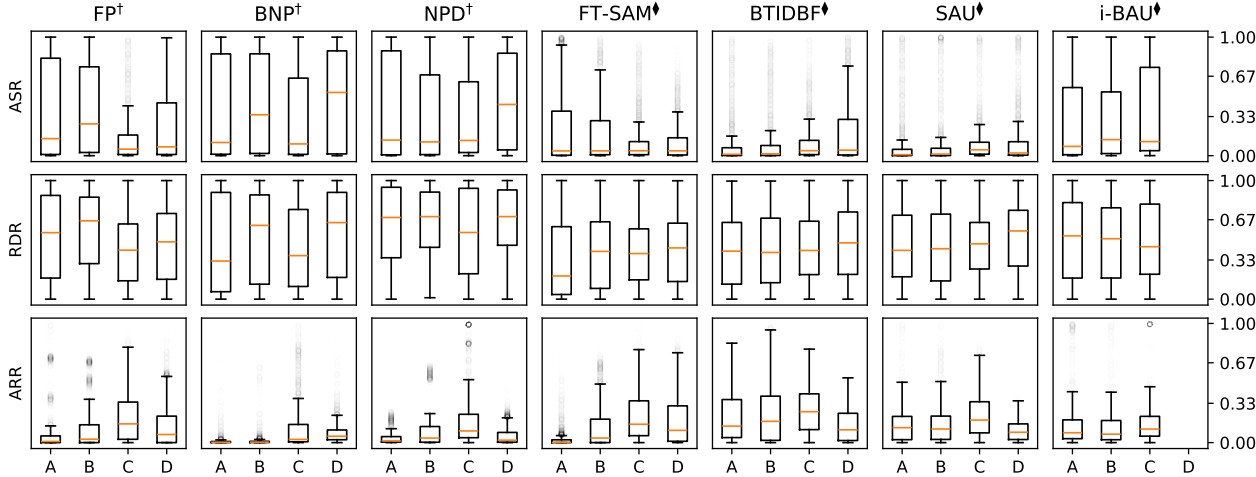

Figure 8: Box plots of the ASR, RDR, and ARR results for the selected approaches and different model architectures. † = Pruning and ♦ = Fine-tuning. A = VGG, B = ResNet, C = EfficientNet and D = MobileNet. Note: i-BAU results for MobileNet are not shown because its implementation is incompatible with this architecture.

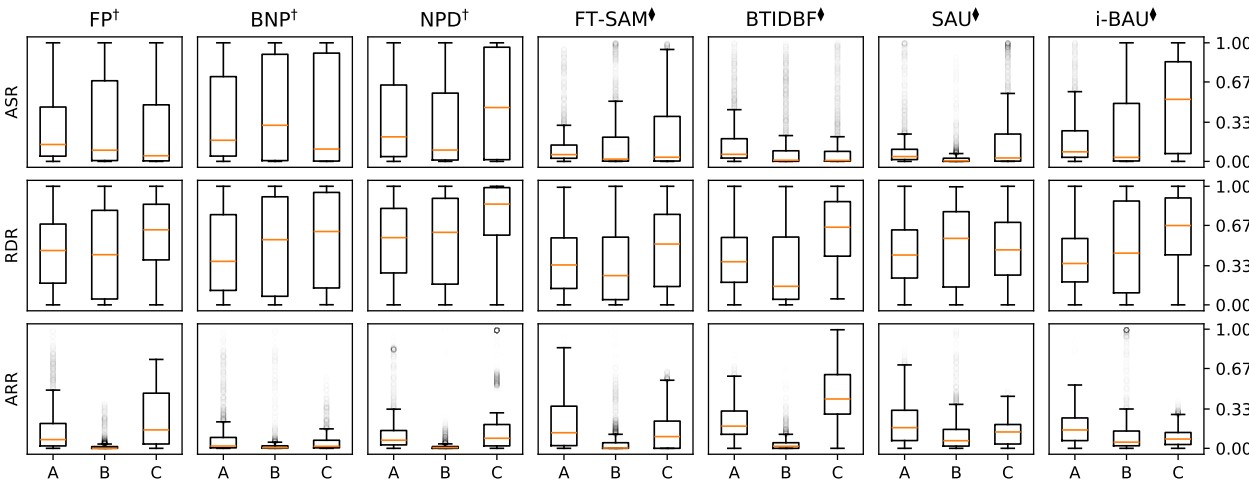

Figure 9: Box plots of the ASR, RDR, and ARR results for the selected approaches and different datasets. † = Pruning and ♦ = Fine-tuning. A = CIFAR-10, B = GTSRB and C = Tiny-ImageNet.

Except for BNP, each approach exhibits variation in the median or tail of the ARR distribution across datasets. In particular, FP and NPD show significant variation in tail weight for CIFAR-10 and Tiny-ImageNet. For RDR, increasing the complexity of the classification task generally leads to worse performance, with SAU being the exception. This is evident from the increase in median RDR from CIFAR-10 to GTSRB and from GTSRB to Tiny-ImageNet. A similar trend is present for ASR, where NPD, FT-SAM, SAU and i-BAU perform worse as task complexity increases.

# 7 Discussion

In this section, we synthesize the major findings of our survey by connecting the literature reviewed in Sections 3 and 4 with the evaluation results in Section 6. We first identify four high-level failure modes that recur across method categories (Section 7.1), then ground these in a per-method analysis linking design-time

assumptions to empirical shortcomings (Section 7.2), discuss broader methodological implications for the field (Section 7.3), and outline open challenges in Section 8.

## 7.1 Common Failure Modes Across Method Categories

Despite the diversity of the surveyed methods, four failure modes recur across categories.

**F1. Overfitting under limited mitigation data:** Fine-tuning methods deteriorate sharply when $\mathcal{D}_m$ contains few samples per class, whereas the discrete nature of pruning provides an implicit regularization that limits the impact of low SPC. The most affected methods are those whose objectives operate directly on $\mathcal{D}_m$ via optimization against a perturbed copy or via a sharpness-aware step.

**F2. Assumption-driven inconsistency:** The empirical shortcomings observed for each evaluated method are consistent with the assumption it makes about backdoor behavior. Methods built around a particular trigger geometry, and methods built around a single observable feature of backdoored models, both show inconsistency precisely on the dimensions of variation (attack, dataset, architecture) over which their assumptions were not validated.

**F3. RDR collapse:** Across every category, methods that successfully reduce ASR fail to comparably restore the correct classification of backdoor samples. When evaluations report ASR without RDR, a low-ASR/high-RDR method can be mistaken for a successful defense even though triggered images remain misclassified, just into different incorrect classes. The recurrence of this pattern across methods with otherwise distinct mechanisms is striking, and Section 7.2 returns to a possible structural explanation rooted in how the surveyed mitigation objectives are constructed.

**F4. Hyperparameter brittleness:** Most evaluated methods include at least one hyperparameter that significantly affects performance (Supplementary Materials, Table I). Validation-based hyperparameter selection is impractical for mitigation because ASR and RDR are not observable to the defender, so a validation set can only inform ARR. Methods that report few hyperparameters but require careful tuning are therefore likely to be less practical than their original evaluations suggest.

## 7.2 From Method Assumptions to Empirical Shortcomings

Table 5 summarizes key assumptions of each evaluated method alongside the most directly relevant shortcoming observed in our evaluation, organized by the method categories of Sections 3 and 4. We emphasize that the mappings in Table 5 are interpretive rather than mechanistic: many shortcomings have multiple plausible causes, and we draw the connection only where the empirical evidence in our evaluation is consistent with the design-time assumption. Several patterns are visible when the table is read across the method-category groupings; most are localized to particular categories, while one cuts across all of them.

Trigger-geometry assumptions cross-cut the categorization. The $\ell_1$-sparse assumption sits within Synthesis Unlearn (NC, MESA, BAERASER); the $\ell_2$-bounded assumption appears in both Additive Pruning (NPD) and Adversarial Training (PBE, i-BAU, SAU). The observed shortcomings of both norm-bound assumptions are similar, inconsistent ASR reduction across attack types and limited improvement in RDR, suggesting that the choice of norm constraint is not the bottleneck for trigger-modeling methods, and that refining the norm in isolation is unlikely to yield a substantive improvement. Within this group, several methods make additional method-specific assumptions: NPD relies on the second-largest logit as a proxy for the backdoor target; PBE relies on a directional bias of PGD adversarial examples; SAU relies on filtering candidate triggers through the original $\theta$. Each of these heuristics introduces its own observed shortcoming (Table 5), indicating that improving trigger-modeling will require addressing both the norm constraint *and* the target-class / adversarial-example heuristic.

Within pruning, the evaluated methods rely heavily on a single observable feature of backdoored models: batch-normalization statistics for BNP, spectral norms for CLP, individual filter activations for FMP, and channel activation magnitudes for MM-BD / UNIT. In each case, the corresponding empirical shortcoming is

variation across attacks or architectures, which is consistent with the observable feature not capturing backdoor behavior universally. Sensitivity to mitigation-data availability is also concentrated around particular assumption types rather than around one method category. ANP / AWM in Masking-based Pruning, PBE in Adversarial Training, and FT-SAM in Conventional Fine-Tuning all deteriorate when SPC is reduced (F1), reflecting their shared reliance on the structure of $\mathcal{D}_m$ during optimization.

Cutting across all method categories, the RDR-related entries in Table 5 share a common surface feature: methods that substantially reduce ASR do not exhibit comparable RDR improvements, despite their otherwise distinct mechanisms. In the case of methods that use adversarial examples, mitigation losses typically include a term that penalizes classification rather than a term that explicitly rewards classification of those samples as their original label. Concretely, SAU's outer objective and NPD's adversarial term both push the model away from predicting the estimated target class on adversarial samples but include no corresponding term rewarding prediction of the correct class. MM-BD and UNIT cap activation magnitudes without any recovery-aligned term. Finally, most pruning objectives select components using proxies aligned with reducing the attack success rate rather than with recovering correct classification. Under this kind of objective, correcting classification is only an implicit objective, which can reduce ASR without producing high RA.

Together, these patterns reinforce the aggregate finding of Section 6.1.4: progress in mitigation has been highly localized. New methods tend to refine an assumption that already exists in the literature rather than identify a universal characteristic of backdoor behavior. The absence of such a universal characteristic across our results suggests that future work may need to combine multiple complementary assumptions, or to reformulate the mitigation task in terms that do not rely on a single observable property of backdoored models.

### 7.3 Implications for the Field

Beyond the per-method shortcomings discussed in Section 7.2, our review highlights two structural issues distinct from any individual method: one in how the observations underpinning current methods are validated, and one in how mitigation success itself is measured.

The design-time observations underpinning many methods are validated on a narrow set of attacks, datasets, and architectures. Examples include CLP's analysis of the relationship between UCLC and TAC on a single dataset and architecture, and BNP's pre-activation distributions illustrated for a single filter. Our results indicate that none of these observations can be deemed universal characteristics of backdoor behavior; rather, they are indicative of backdoor behavior under the specific settings tested by each proposal. The field would benefit from a stronger evaluation norm that requires validation of each motivating observation across multiple datasets and architectures before that observation is used to ground a method.

Turning from how methods are designed to how they are evaluated, the mitigation problem is widely treated as the problem of reducing ASR, with RDR (and RA) reported as secondary or omitted entirely. Because ASR+RA $\leq 1$ with RA $\leq$ ACC in typical targeted-attack evaluations, a method can reduce ASR by inducing misclassification into any non-target class, which provides no practical benefit. RDR captures whether the original classification is genuinely restored and should be reported as a first-class metric alongside ASR. This concern is reinforced by the cross-cutting structural observation in Section 7.2: most surveyed objectives include explicit terms penalizing classification as the (estimated) target but few include terms rewarding classification as the original class, which is consistent with the RDR-collapse pattern (F3) seen across method categories. While we are careful not to claim that objective design is the sole or even dominant cause of this pattern, it is a candidate factor that, unlike many of the per-method assumptions surveyed, can be addressed without making further assumptions about the trigger or the backdoored model.

## 8 Conclusion

We presented a focused survey and benchmark of 19 backdoor-mitigation methods for image recognition, spanning eight attacks, four model architectures, three datasets, three poisoning ratios, and three data-availability settings, over 120,000 individual experiments in total. Our analysis combines a per-method theoretical review with an aggregated empirical comparison against the seminal 2018 baselines FP and FT.

From our theoretical and empirical analysis, we synthesize four main takeaways:

- Despite a steady stream of new mitigation proposals, none consistently outperforms FP across ASR, RDR, and ARR jointly (Section 6.1.4). Among pruning methods, ANP, BNP, CLP, and NPD perform comparably to FP but do not improve upon it. Among fine-tuning methods, FT-SAM, SAU, BTI-DBF, and i-BAU outperform FT on ASR but do not consistently improve RDR.

- Four failure modes recur across categories (Section 7.1): overfitting under limited mitigation data, assumption-driven inconsistency, RDR collapse, and hyperparameter brittleness. The empirical shortcomings of each method are consistent with the assumptions made about backdoor behavior in each work (Section 7.2), suggesting that the observed inconsistencies are tied to design-time choices rather than being only random variation.

- Trigger-modeling methods, used in nearly half of the surveyed works, are among the strongest performers in our evaluation (i.e., SAU, i-BAU, and BTI-DBF). Even so, these methods are not without issues that require careful consideration in future work. Both $\ell_1$- and $\ell_2$-bounded modeling exhibit similar shortcomings, inconsistent ASR across attack types and limited RDR improvement, suggesting that the choice of norm constraint is not the limiting factor and that further progress will require careful reformulation of trigger-modeling rather than refinement of the norm alone.

- RDR remains the most under-served metric in the field. Methods that reduce ASR rarely restore correct classification, leaving backdoor images misclassified into different incorrect classes rather than recovered to their original label. While we do not attempt to attribute this to any single mechanism in any individual method, the recurrence of the pattern across otherwise distinct methods is consistent with a structural feature of the surveyed objectives, namely that they predominantly penalize classification as the target without explicitly rewarding classification as the original class.

These takeaways translate into the following open challenges for future work:

1. Future fine-tuning methods should explicitly constrain $\theta$, not merely the perturbation space or the optimization objective, to address the bias-variance trade-off in low-data regimes (F1).

2. Because ASR and RDR are not observable to the defender, the field needs hyperparameter-selection procedures that rely only on $\mathcal{D}_m$. Promising directions include uncertainty or activation-distribution heuristics that do not require backdoor samples (F4).

3. RDR should be a first-class evaluation metric in future work. We recommend that new methods report ASR *and* RDR jointly, and report variance, not only median performance (F3). Future work should also directly study how RDR gains can be made, even at the cost of some ASR performance.

4. The patterns visible in Table 5 suggest that no single assumption, whether about trigger geometry or about an observable feature of backdoored models, yields consistent mitigation. Future methods should consider combining complementary assumptions or developing more fundamental mechanisms that better describe the behavior of backdoored models (F2). The community should carefully validate future proposals across diverse architectures, including ViT-style models, before claiming "universal" properties of backdoored models.

In summary, while measurable progress has been made on ASR reduction, the practical robustness, recovery capability, and generalizability of mitigation methods remain open. We hope the unified benchmark and the failure-mode synthesis presented here will help focus future research on the gaps that matter most for real-world deployment.

# 9 Acknowledgment

Dimity Miller acknowledges ongoing support from the QUT Centre for Robotics. We acknowledge the support of QUT eResearch for providing the computing facilities required to run our experiments.

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

# A    Appendix

# B    Experimental Parameters

# C   Complete Results

## C.1   Data Availability

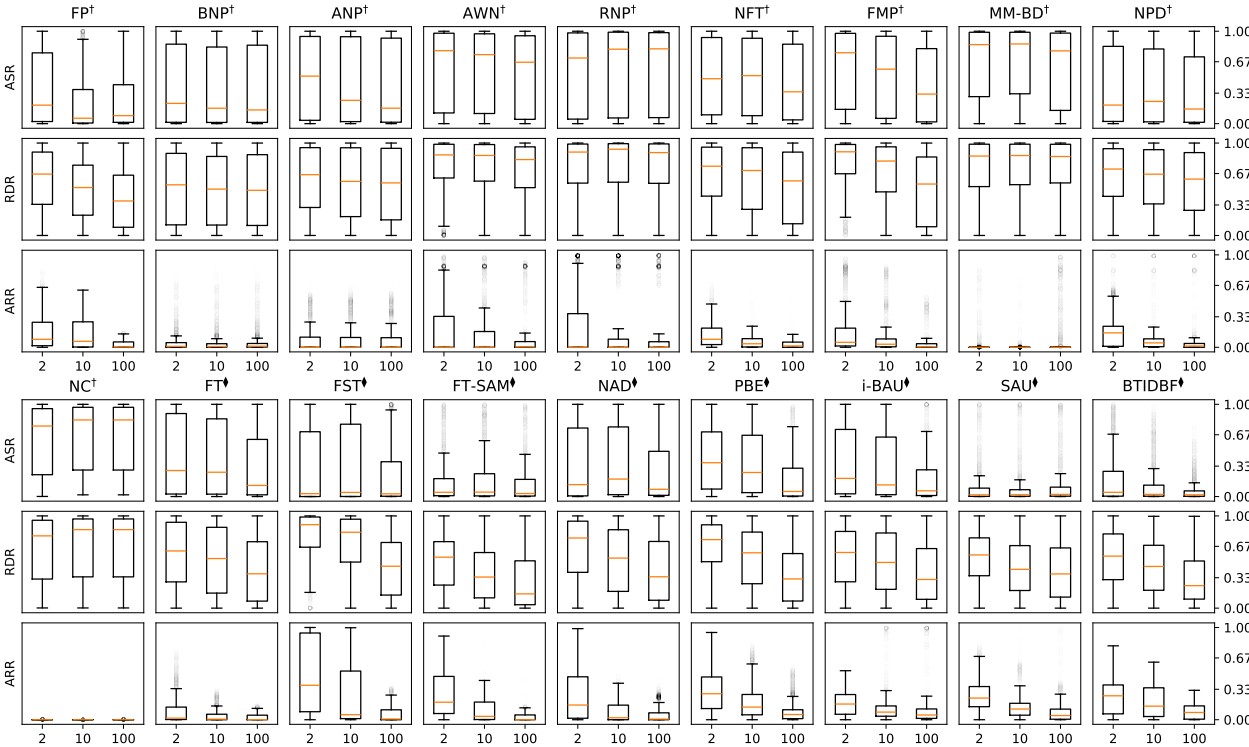

Figure 10: Box plots of the ASR, RDR, and ARR results for each approach across all considered scenarios. † = Pruning and ◊ = Fine-tuning. Note: NC results are only for CIFAR-10.

## C.2 Backdoor Attack

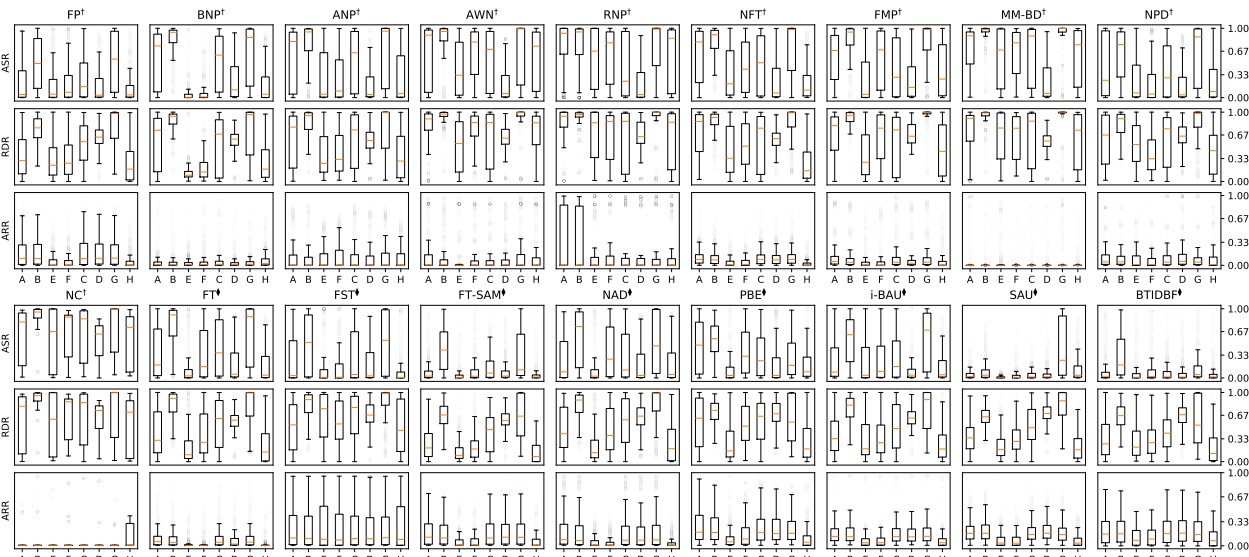

Figure 11: Box plots of the ASR, RDR, and ARR results for each approach and different attack types. † = Pruning and ◇ = Fine-tuning. A = BadNet, B = Blended, C = LF, D = Signal, E = BPP, F = IAB, G = SSBA and H = WaNet. Note: NC results are only for CIFAR-10.

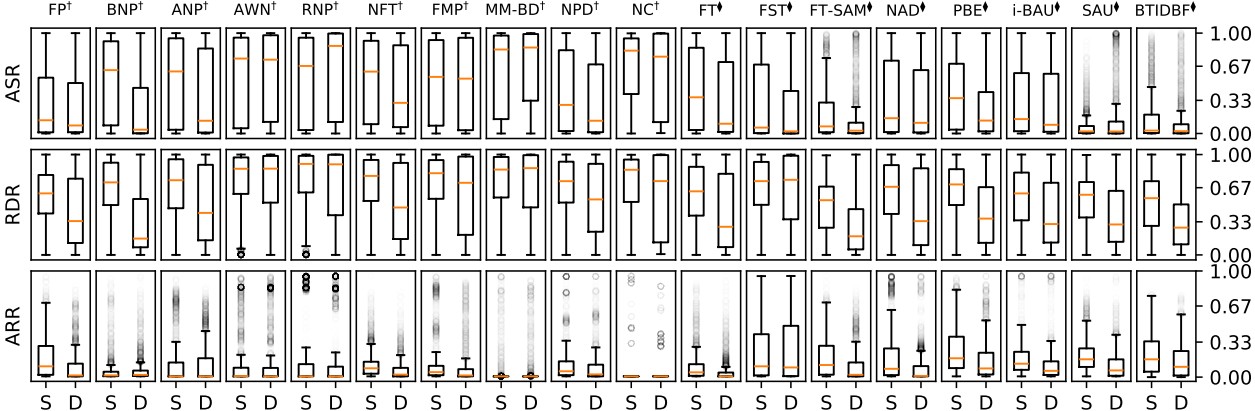

Figure 12: Box plots of the ASR, RDR, and ARR results for each approach and both static and dynamic attacks. † = Pruning and ◇ = Fine-tuning. S = Static and D = Dynamic. Note: NC results are only for CIFAR-10.

## C.3 Model Architecture

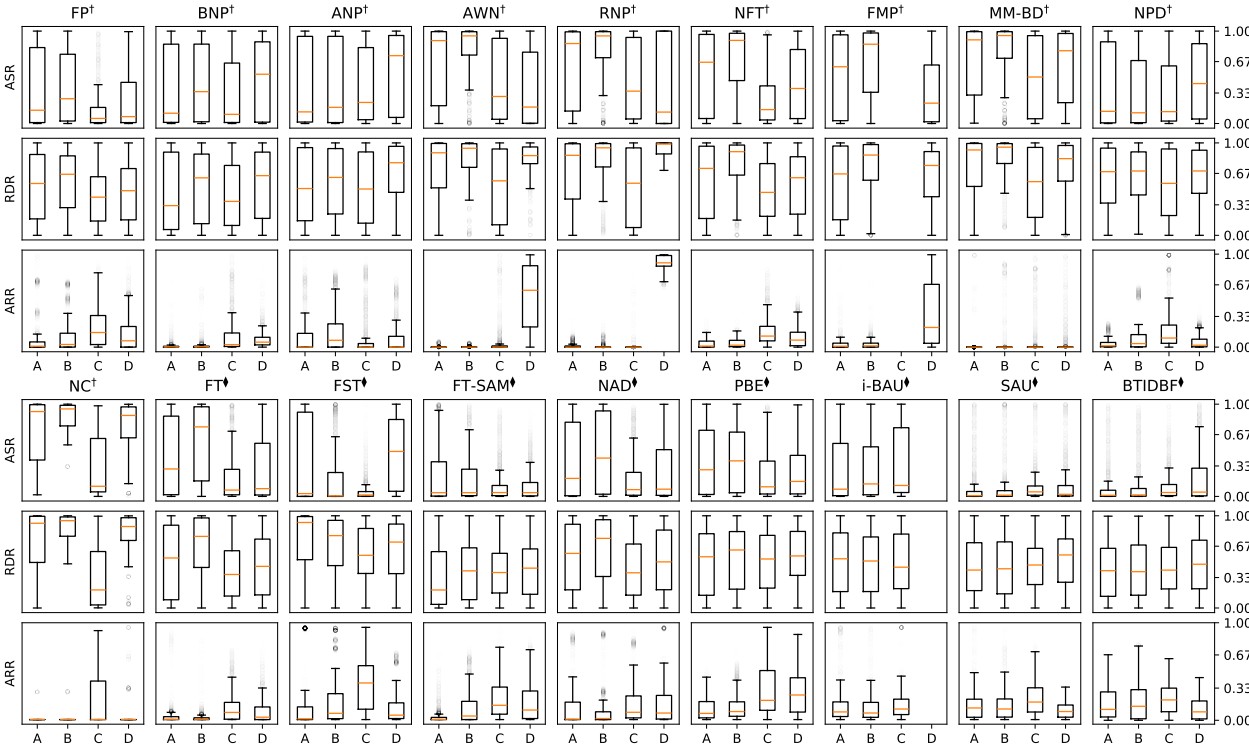

Figure 13: Box plots of the ASR, RDR, and ARR results for each approach and different model architectures. † = Pruning and ◇ = Fine-tuning. A = VGG, B = ResNet, C = EfficientNet and D = MobileNet. Note: The current i-BAU implementation is incompatible with the MobileNet architecture and NC results are only for CIFAR-10.

## C.4   Dataset

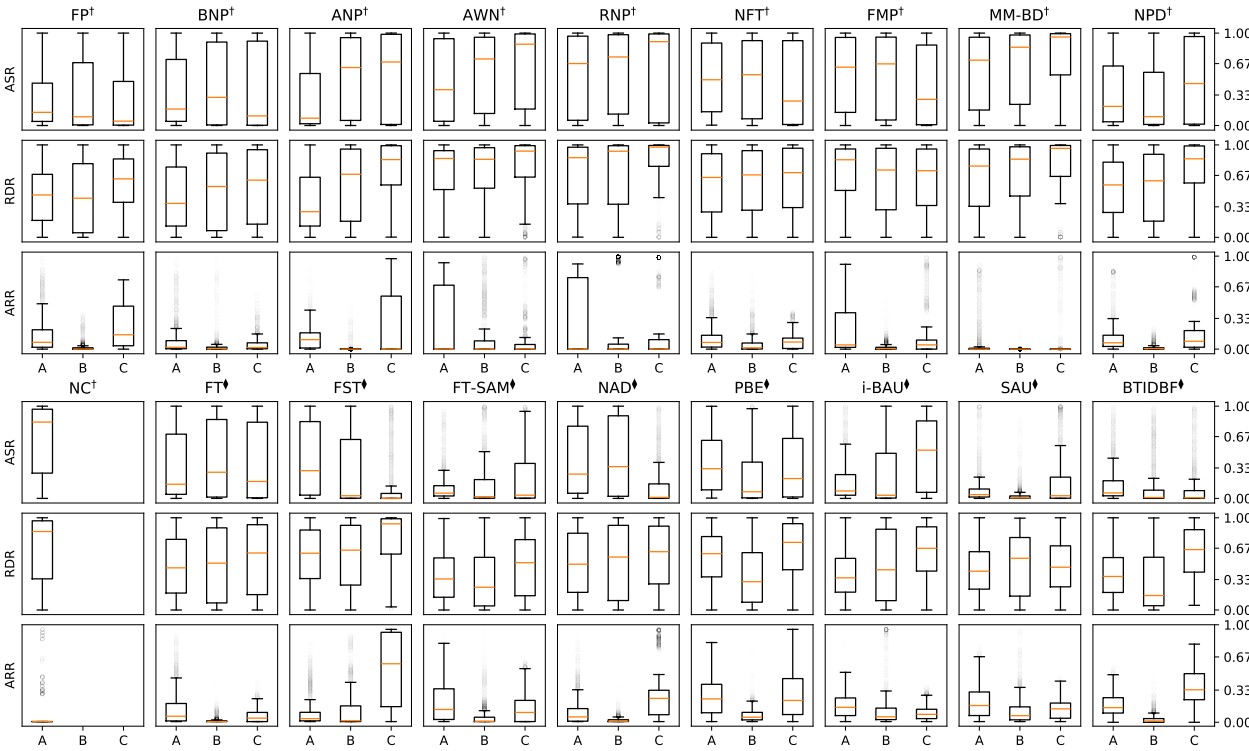

Figure 14: Box plots of the ASR, RDR, and ARR results for the selected approaches and different datasets. † = Pruning and ◇ = Fine-tuning. A = CIFAR-10, B = GTSRB and C = Tiny-ImageNet. Note: NC results are only for CIFAR-10.

## C.5 Poisoning Ratio

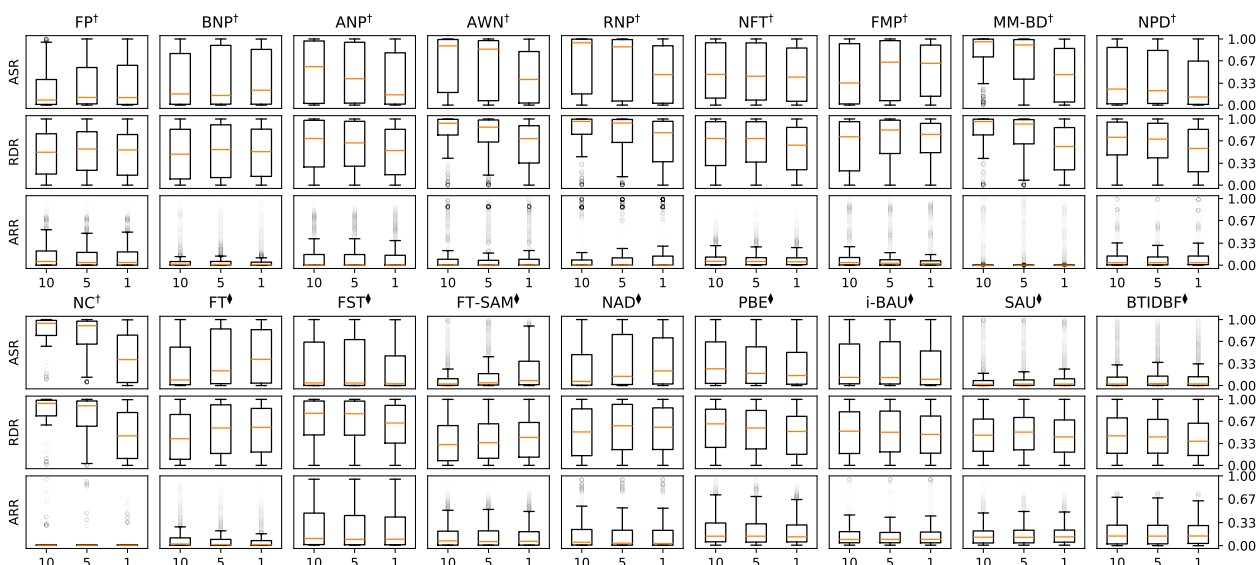

Figure 15: Box plots of the ASR, RDR, and ARR results for the selected approaches and different poisoning ratios (%). † = Pruning and ◇ = Fine-tuning. Note: NC results are only for CIFAR-10.

Table 5: Mapping from key method assumptions to the general empirical shortcomings observed in our evaluation. Rows are grouped by the method categories introduced in Sections 3 and 4. Methods relying on multiple distinct assumptions (e.g., NPD, PBE, SAU) appear once per assumption. Failure-mode tags (F1–F4) refer to the failure modes introduced in Section 7.1; in particular, several rows are tagged with F3 to highlight the cross-cutting RDR-collapse pattern discussed in the prose below.

| Method | Key Assumption | Observed Empirical Shortcoming |
|---|---|---|
| *Metric-based Pruning* | | |
| **BNP** | Pre-activation values follow a Gaussian distribution; backdoor filters are separable via BN-statistic divergence. | ASR distribution exhibits a heavy tail across attack types, suggesting that the assumed pre-activation distribution does not hold universally. |
| **CLP** | Spectral upper bound (UCLC) correlates with backdoor sensitivity (TAC) across all architectures. | ARR varies substantially across architectures, indicating that the UCLC–TAC relationship is not architecture-invariant. |
| **FMP** | Each individual filter associated with the backdoor can independently induce backdoor behavior. | Inconsistent ASR across attacks, consistent with backdoor functionality being distributed across filters rather than localized to independent ones. |
| *Masking-based Pruning* | | |
| **ANP / AWM** | Weight or input perturbations maximizing loss specifically target backdoor components. | Performance deteriorates substantially as mitigation-data availability is reduced (F1), suggesting that the perturbation–backdoor coupling is not robust under low SPC. |
| **NFT** | Trigger can be modeled as a small-magnitude perturbation on the data manifold (MixUp upper bound). | Variable ASR across attack types, with weaker performance on attacks whose triggers are not well-approximated by small on-manifold perturbations. |
| *Additive Pruning* | | |
| **MM-BD / UNIT** | Upper-bounding channel activations is sufficient to recover correct classification of $\mathcal{D}_b$. | Reductions in ASR are not accompanied by comparable improvements in RDR, indicating that activation bounding alone does not restore correct classification (F3). |
| **NPD** | Trigger is $\ell_2$-bounded (shared with PBE / i-BAU / SAU under Adversarial Training). | $\ell_2$-bounded modeling does not consistently outperform $\ell_1$-based approaches across attack types and provides limited improvement on RDR (F3). |
| **NPD** | Second-largest logit corresponds to the backdoor target. | ASR variance increases with task complexity, consistent with the second-largest logit being a less reliable proxy for the backdoor target in higher-class-count settings. |
| *Conventional Fine-Tuning* | | |
| **FT-SAM** | Sharp local minima are the primary cause of fine-tuning's failure on backdoors. | Performance is highly sensitive to mitigation-data availability (F1), and gains on ASR are not matched by gains on RDR (F3). |
| *Synthesis Unlearn* | | |
| **NC / MESA / BAERASER** | Trigger is sparse ($\ell_1$-bounded). | $\ell_1$-bounded trigger inversion does not yield consistent ASR reduction across attack types, particularly where triggers are not sparse. |
| *Adversarial Training* | | |
| **PBE / i-BAU / SAU** | Trigger is $\ell_2$-bounded (shared with NPD under Additive Pruning). | $\ell_2$-bounded modeling does not consistently outperform $\ell_1$-based approaches across attack types and provides limited improvement on RDR (F3). |
| **PBE** | PGD-generated adversarial examples are biased toward the backdoor target class. | Performance varies across settings, suggesting that the assumed bias of PGD-generated examples toward the target is not consistent. |
| **SAU** | Filtering candidate triggers through original $\theta$ distinguishes them from adversarial examples. | Robust ASR reduction is achieved, but RDR remains highly variable – distinguishing triggers from adversarial examples does not, on its own, recover correct classification (F3). |

Table 6: Experimental parameters used by each approach. AR = Accuracy Ratio, UT = Unlearn Threshold, RD = Recovery Drop

| Reference | Approach | Implementation | Training Parameters | | | Hyperparameters | | |
|---|---|---|---|---|---|---|---|---|
| | | | CIFAR-10 | GTSRB | Tiny | CIFAR-10 | GTSRB | Tiny |
| Liu et al. (2018) | FP | BackdoorBench | AR = 0.1 | AR = 0.1 | AR = 0.1 | N/A | N/A | N/A |
| Zheng et al. (2022a) | BNP | BackdoorBench | N/A | N/A | N/A | $\lambda = 3$ | $\lambda = 3$ | $\lambda = 3$ |
| Zheng et al. (2022b) | CLP | BackdoorBench | N/A | N/A | N/A | $\lambda = 3$ | $\lambda = 3$ | $\lambda = 3$ |
| Wu & Wang (2021) | ANP | BackdoorBench | AR = 0.1 | AR = 0.1 | AR = 0.1 | $\epsilon = 0.4$ $\lambda = 0.2$ | $\epsilon = 0.4$ $\lambda = 0.2$ | $\epsilon = 0.4$ $\lambda = 0.2$ |
| Chai & Chen (2022) | AWM | GitHub | N/A | N/A | N/A | $\lambda_1 = 0.9$ $\lambda_2 = 0.1$ $\lambda_3 = 10^{-7}$ | $\lambda_1 = 0.9$ $\lambda_2 = 0.1$ $\lambda_3 = 10^{-7}$ | $\lambda_1 = 0.9$ $\lambda_2 = 0.1$ $\lambda_3 = 10^{-7}$ |
| Li et al. (2023) | RNP | GitHub | UT = 0.1 RD = 0.02 | UT = 0.1 RD = 0.02 | UT = 0.1 RD = 0.02 | N/A | N/A | N/A |
| Karim et al. (2024) | NFT | GitHub | $\beta = 0.5$ $\alpha = 0.8$ | $\beta = 0.5$ $\alpha = 0.8$ | $\beta = 0.5$ $\alpha = 0.8$ | $\lambda = 0.001$ | $\lambda = 0.001$ | $\lambda = 0.001$ |
| Huang & Bu (2023) | FMP | GitHub | N/A | N/A | N/A | N/A | N/A | N/A |
| Wang et al. (2023) | MM-BD | GitHub | AR = 0.05 $\alpha = 1.2$ | AR = 0.05 $\alpha = 1.2$ | AR = 0.05 $\alpha = 1.2$ | $\lambda = 0.5$ | $\lambda = 0.5$ | $\lambda = 0.5$ |
| Zhu et al. (2024b) | NPD | BackdoorBench | N/A | N/A | N/A | $\lambda_1 = 1$ $\lambda_2 = 0.4$ $\lambda_3 = 0.4$ | $\lambda_1 = 1$ $\lambda_2 = 0.5$ $\lambda_3 = 0.5$ | $\lambda_1 = 1$ $\lambda_2 = 0.4$ $\lambda_3 = 0.4$ |
| Wang et al. (2019a) | NC | BackdoorBench | N/A | N/A | N/A | $\lambda = 10^{-3}$ | $\lambda = 10^{-3}$ | $\lambda = 10^{-3}$ |
| Liu et al. (2018) | FT | BackdoorBench | N/A | N/A | N/A | N/A | N/A | N/A |
| Min et al. (2024) | FST | GitHub | N/A | N/A | N/A | $\lambda = 0.2$ | $\lambda = 0.01$ | $\lambda = 0.001$ |
| Zhu et al. (2023) | FT-SAM | BackdoorBench | N/A | N/A | N/A | N/A | N/A | N/A |
| Li et al. (2021a) | NAD | BackdoorBench | N/A | N/A | N/A | $\lambda \in \{500, 1000\}$ | $\lambda \in \{500, 1000\}$ | $\lambda \in \{500, 1000\}$ |
| Mu et al. (2023) | PBE | GitHub | N/A | N/A | N/A | N/A | N/A | N/A |
| Zeng et al. (2022) | i-BAU | BackdoorBench | N/A | N/A | N/A | N/A | N/A | N/A |
| Wei et al. (2023) | SAU | BackdoorBench | N/A | N/A | N/A | $\lambda_1 = 1$ $\lambda_2 = 0$ $\lambda_3 = 1$ | $\lambda_1 = 1$ $\lambda_2 = 0$ $\lambda_3 = 1$ | $\lambda_1 = 1$ $\lambda_2 = 0$ $\lambda_3 = 1$ |
| Xu et al. (2024) | BTI-DBF | GitHub | N/A | N/A | N/A | N/A | N/A | N/A |

