# OpenReview forum: "Countering Backdoor Attacks in Image Recognition: A Survey and Evaluation of Mitigation Strategies"
_TMLR — Accepted by TMLR_

### Review · Reviewer_Bwca · 2026-04-09

**Summary Of Contributions:**

This paper presents a systematic survey and large-scale empirical evaluation of backdoor mitigation methods for image classification. The authors organize prior work into pruning-based and fine-tuning-based approaches, further discuss their assumptions and limitations, and benchmark 16 representative methods under a unified experimental framework. The evaluation spans 8 backdoor attacks, 3 datasets, 4 model architectures, multiple poisoning ratios, and different data-availability settings, resulting in more than 122,000 experiments. The main contribution is not only the breadth of the survey, but also the finding that many recently proposed methods do not consistently outperform early baselines such as FP and FT, and that lowering ASR alone is insufficient if recovery performance, captured by RDR, remains poor.

**Audience:**

Yes

**Audience Explanation:**

I believe this paper would be of clear interest to a meaningful portion of the TMLR audience, especially researchers working on machine learning security, adversarial robustness, trustworthy AI, and the deployment of neural systems in safety- or security-sensitive settings. Backdoor defenses remain an active and practically important topic, and this paper offers value both as a structured survey and as a large-scale reassessment of the field under a unified benchmark. The findings are particularly useful because they challenge an implicit narrative of steady progress by showing that many newer methods are not consistently better than classical baselines, and because they emphasize that recovery-oriented evaluation should matter more in future work. These conclusions are relevant not only to those designing new defenses, but also to readers who rely on published claims when choosing methods for practical use.

**Broader Impact Concerns:**

The overall impact of this work is likely positive. The paper is focused on understanding and improving defenses against backdoor attacks, which is directly relevant to the security and trustworthiness of machine learning systems. As with much work in ML security, there is a dual-use aspect because the paper discusses attack models, trigger properties, and the limitations of existing defenses. However, these attack families are already well established in the literature, and the main contribution here is evaluative and defensive rather than offensive. I therefore do not see ethical concerns beyond the standard dual-use considerations that apply across this area. On balance, the paper’s likely effect is to encourage more realistic evaluation practices and more robust future defense design.

**Claims And Evidence:**

Yes

**Claims Explanation:**

The paper’s central claims are broadly supported by substantial empirical evidence. The experimental evaluation is extensive, covers many relevant dimensions, and is presented in a way that makes the overall trends visible, especially the instability of many methods across attacks, datasets, architectures, and data regimes. The claims that many recent approaches do not reliably surpass older baselines, and that RDR is an important but underemphasized metric, are generally consistent with the reported results. That said, the evidential strength is weakened somewhat by clarity and presentation issues. The exact scope of what is surveyed versus what is fully benchmarked is not always stated with complete consistency, and there are places where the framing of the contribution could be tighter. In addition, while the visual summaries are useful, the paper relies mainly on descriptive comparisons rather than more formal statistical analysis. I also think the discussion of why current methods often reduce ASR without meaningfully restoring correct classification could be developed more deeply. Overall, the evidence is meaningful and largely convincing, but not as clear or airtight as it could be.

**Requested Changes:**

The paper has clear potential, but several revisions are needed to strengthen it. First, the manuscript should be carefully cleaned up for consistency and completeness, including fixing unresolved placeholders such as “X individual experiments” and ensuring that the abstract, tables, and experimental sections all align on how many methods are surveyed, benchmarked, excluded, or only partially evaluated. Second, the authors should make the main takeaways more prominent earlier in the paper, especially the result that many recent methods do not consistently improve over FP and FT, and the argument that low ASR without strong recovery is not sufficient. Third, the writing would benefit from a thorough language edit to address grammar, phrasing, and naming inconsistencies, which currently make the paper feel less polished than the underlying work deserves. Fourth, I would encourage the authors to deepen the analysis of RDR and explain more clearly why existing defenses often succeed at suppressing target-class misclassification without restoring correct classification of triggered samples. Finally, adding some stronger statistical support for comparative claims would improve the rigor of the evaluation. The first three of these are close to critical for my recommendation, while the latter two would significantly strengthen the paper.

---

> ### Author Response · Authors · 2026-05-15
>
> We thank the reviewer for the detailed and constructive review, and in particular for identifying manuscript-level issues such as placeholders, naming inconsistencies, and misalignment among the abstract, tables, and experimental sections. The revision addresses these issues explicitly.
>
> **Comment 3.1:** *The manuscript should be carefully cleaned up for consistency and completeness, including fixing unresolved placeholders such as "X individual experiments" and ensuring that the abstract, tables, and experimental sections all align on how many methods are surveyed, benchmarked, excluded, or only partially evaluated.*
>
> **Response:** We have addressed this throughout the manuscript. The "X individual experiments" placeholder has been removed; the abstract, contribution list in Section 1, and experimental description in Section 5 now consistently state "over 120,000 individual experiments." We have also reconciled the method counts: Sections 3–4 review 23 mitigation methods, 19 of which are benchmarked in Section 5. Table 1 reports the benchmarked-method count, consistent with its caption.
>
> **Comment 3.2:** *The authors should make the main takeaways more prominent earlier in the paper, especially the result that many recent methods do not consistently improve over FP and FT, and the argument that low ASR without strong recovery is not sufficient.*
>
> **Response:** We have made several changes to surface the main takeaways earlier. The abstract now states the central finding directly. The contribution list in Section 1 includes the FP/FT comparison as a dedicated bullet. The introductory text also flags the ASR–RA distinction and explains why low ASR alone is insufficient. Within the results, Section 6.1.4 now leads with the FP/FT-rectangle framing of Fig. 5. The Conclusion then opens with these points as the first two takeaways.
>
> **Comment 3.3:** *The writing would benefit from a thorough language edit to address grammar, phrasing, and naming inconsistencies, which currently make the paper feel less polished than the underlying work deserves.*
>
> **Response:** We have conducted a full language and consistency pass. We corrected naming and category inconsistencies, removed unresolved placeholders, tightened the abstract and contribution list, and revised the openings of Sections 6 and 7 to improve readability and flow.
>
> **Comment 3.4:** *Deepen the analysis of RDR and explain more clearly why existing defenses often succeed at suppressing target-class misclassification without restoring correct classification of triggered samples.*
>
> **Response:** We have expanded the RDR analysis and now treat it as a central topic rather than a secondary observation. Three changes are particularly relevant:
>
> - Section 7.1 identifies "RDR collapse" as Failure Mode F3, separate from the broader issue of method inconsistency.
> - Section 7.2 connects the RDR-collapse pattern to a cross-cutting structural feature of the surveyed objectives: many methods include terms that penalize classification as the target class, but few include terms that explicitly reward recovery of the original class.
> - Section 7.3 states the methodological implication more directly: future work should report ASR and RDR jointly, treat RDR as a first-class metric, and report variance alongside median performance. The third open challenge in the Conclusion makes this recommendation explicit and calls for direct study of how RDR gains can be achieved, even when this requires trading off some ASR reduction.
>
> **Comment 3.5:** *Adding some stronger statistical support for comparative claims would improve the rigor of the evaluation.*
>
> **Response:** We thank the reviewer for this suggestion. After careful consideration, we have not introduced formal null-hypothesis tests, and we now explain this choice more transparently.
>
> The structure of our evaluation does not naturally fit a single null-hypothesis significance-testing framework. The main claims are not simple central-tendency comparisons for which one $p$-value would be most informative. Instead, they concern dispersion across a large experimental grid, heavy tails, instability across attack types and architectures, and whether any method enters the FP/FT baseline rectangles on all three metrics simultaneously. Collapsing this heterogeneous grid into a single test could obscure the practical behavior that the benchmark is intended to reveal. We have therefore strengthened the distributional comparisons, baseline-rectangle analysis, and wording around method superiority while avoiding claims that exceed the evidence. In particular, we use "consistently improves" rather than "significantly improves" where the point is robustness across settings rather than formal statistical significance.

---

### Review · Reviewer_H874 · 2026-04-14

**Summary Of Contributions:**

This paper surveys methods for mitigating backdoor attacks in image recognition. The authors categorize existing defense approaches into two main families: model-pruning-based methods and fine-tuning-based methods. In addition, they evaluate 16 representative backdoor defense methods against various backdoor attacks on CIFAR-10 under a unified experimental setting.

**Audience:**

Yes

**Audience Explanation:**

1. I appreciate that the authors benchmark and compare different backdoor defenses under the same experimental setting. This is particularly important for the trustworthy AI community, as it provides a more reliable view of the current progress and practical effectiveness of existing backdoor defense methods.

2. The conclusions drawn from the experimental results are useful. In particular, the paper highlights that many methods have unstable performance across different attacks and settings, and that recovering correct predictions on triggered inputs remains a major open challenge. These observations can help guide future research in the trustworthy AI community.

**Claims And Evidence:**

Yes

**Claims Explanation:**

1. This paper is well written and easy to follow. The authors organize existing backdoor defense approaches into two main categories, model-pruning-based methods and fine-tuning-based methods, and discuss representative approaches in each family in detail. I believe this survey provides a clear overview of the current landscape of backdoor defenses for image recognition, especially for readers without a strong background in ML backdoor attacks or trustworthy AI.

**Requested Changes:**

1. A major limitation of this paper is that all cited works appear to be from 2024 or earlier. I suggest that the authors cite and discuss more recent papers on backdoor attacks and defenses, such as [r1, r2, r3].

**References**

[r1] Zhu et al. "Towards Sample-specific Backdoor Attack with Clean Labels via Attribute Trigger." TDSC 2025.

[r2] Chen et al. "REFINE: Inversion-Free Backdoor Defense via Model Reprogramming." ICLR 2025.

[r3] Hou et al. "FLARE: Toward Universal Dataset Purification against Backdoor Attacks." TIFS 2025.

---

> ### Author Response · Authors · 2026-05-15
>
> We thank the reviewer for the positive assessment of the survey's accessibility and the value of the unified benchmark, and for highlighting the practical relevance of the findings on instability and recovery.
>
> **Comment 2.1:** *A major limitation of this paper is that all cited works appear to be from 2024 or earlier. I suggest that the authors cite and discuss more recent papers on backdoor attacks and defenses, such as [r1] Zhu et al. "Towards Sample-specific Backdoor Attack with Clean Labels via Attribute Trigger" (TDSC 2025), [r2] Chen et al. "REFINE: Inversion-Free Backdoor Defense via Model Reprogramming" (ICLR 2025), and [r3] Hou et al. "FLARE: Toward Universal Dataset Purification against Backdoor Attacks" (TIFS 2025).*
>
> **Response:** We thank the reviewer for these helpful suggestions. We have updated the manuscript to engage with recent work more clearly.
>
> - **[r2] REFINE (Chen et al., 2025)** and **[r3] FLARE (Hou et al., 2025)** are now cited and discussed in Section 2.3 ("Backdoor Mitigation"). We clarify that both works fall outside the scope of *model mitigation* as defined in our benchmark: FLARE addresses dataset purification by identifying poisoned training samples, whereas REFINE learns an input transformation that removes the backdoor effect without modifying the underlying model. These are valuable defensive contributions, but they require a different evaluation framework from the mitigation setting studied here, where the objective is to remove backdoor behavior from the model while preserving clean-task performance.
> - **[r1] Zhu et al. (2025)** proposes a sample-specific clean-label backdoor attack rather than a mitigation method. We now cite this work in Section 2.2 when discussing the attack-selection scope. We also clarify why the benchmark does not add further specialized attacks: the evaluation is designed around a representative grid of widely studied attacks spanning trigger coverage, consistency, modification mode, and threat model. Adding additional specialized attacks would be valuable future work, but it would change the scope of the controlled benchmark rather than directly addressing the mitigation comparison studied in this paper.

---

> > ### Comment · Reviewer_H874 · 2026-05-22
> >
> > Thanks to the authors for their rebuttal, which addressed most of my concerns. I have no more question and lean to accept this paper.

---

### Review · Reviewer_wNp6 · 2026-04-19

**Summary Of Contributions:**

This paper presents a survey and large-scale empirical evaluation of backdoor mitigation methods for image recognition. It organizes existing approaches into coherent categories, mainly pruning-based and fine-tuning-based methods, and discusses their assumptions and limitations. It further benchmarks 16 mitigation methods across 8 backdoor attacks, multiple datasets, model architectures, and poisoning ratios under a unified setup. The main strength lies in the breadth and consistency of the evaluation, which enables more direct comparison across methods and reveals that many defenses remain unstable across settings, with recent approaches not consistently outperforming earlier baselines. The contribution is centered on systematic synthesis and rigorous empirical analysis rather than introducing new mitigation techniques.

**Audience:**

Yes

**Audience Explanation:**

Backdoor attacks remain an important problem in machine learning security, particularly in scenarios involving third-party data, pre-trained models, or outsourced training. A comprehensive and consistent evaluation of mitigation methods is therefore valuable. The observation that many defenses lack robustness across settings is especially relevant and can help guide future work toward more reliable and generalizable approaches.

**Claims And Evidence:**

Yes

**Claims Explanation:**

The combination of a structured survey and a unified empirical evaluation provides a clearer view of the current landscape, addressing fragmentation in prior work where methods are often evaluated under inconsistent settings. The empirical findings are useful and help highlight important trends, particularly the lack of robustness and consistency across defenses. The paper also draws connections between categories of methods and their underlying assumptions, although these connections are sometimes implicit. To further strengthen its contribution as a survey, the paper could more clearly articulate the key takeaways that emerge from both the survey and the experiments. In particular, explicitly summarizing common failure modes, design trade-offs, and open problems would make the insights more visible and actionable. Some empirical claims would also benefit from deeper analysis or clearer aggregation to better support the conclusions.

**Requested Changes:**

Critical
- Clarify the positioning relative to prior surveys and benchmarks more explicitly in the introduction.
- Strengthen support for key empirical claims, especially comparisons between recent and earlier methods, with clearer aggregation or additional analysis.
- More explicitly synthesize the main insights of the paper, such as common failure modes, trends across method categories, and open challenges.

Strengthen
- Better connect method assumptions to observed failure cases.
- Improve consistency in the level of detail across method descriptions.
- Structure the discussion of limitations more clearly.

---

> ### Author Response · Authors · 2026-05-15
>
> We thank the reviewer for the positive assessment of the breadth and consistency of the evaluation, and for the constructive recommendations on synthesizing the key takeaways and connecting method assumptions to observed failure cases. The revision has been guided directly by these suggestions.
>
> ### Critical Comments
>
> **Comment 1.1:** *Clarify the positioning relative to prior surveys and benchmarks more explicitly in the introduction.*
>
> **Response:** We have revised Section 1 to make the paper's positioning more explicit. Table 1 now includes two additional columns ("Data Avail." and "Multiple Architectures") that contrast the scope of our evaluation with prior surveys and benchmarks. The row for Wu et al. (2022) has also been revised to reflect that, among the nine defensive methods benchmarked in the original paper, only five perform mitigation under the definition used in our work. We have rewritten the surrounding paragraph ("Existing Surveys and Benchmarks") to identify two gaps left by prior work: (i) existing image-domain surveys do not provide method-level depth on assumptions, optimization structure, and resulting limitations; and (ii) practical constraints such as limited mitigation-data availability are not systematically studied. The contribution list now states how our review and benchmark address these gaps.
>
> **Comment 1.2:** *Strengthen support for key empirical claims, especially comparisons between recent and earlier methods, with clearer aggregation or additional analysis.*
>
> **Response:** We have restructured Section 6 to make the FP/FT baseline comparison more explicit. The revised Section 6.1.4 ("Summary and Comparison with Baselines") anchors the central empirical claim in Fig. 5. In that figure, the rectangles defined by the median ARR, RDR, and ASR values of FP and FT mark the regions of joint improvement over the corresponding baseline. The revised text states explicitly that no evaluated pruning method enters FP's rectangle. It also explains that SAU, FT-SAM, i-BAU, and BTI-DBF improve over FT on ASR, but no method improves over FT on all three metrics simultaneously.
>
> **Comment 1.3:** *More explicitly synthesize the main insights of the paper, such as common failure modes, trends across method categories, and open challenges.*
>
> **Response:** We have substantially rewritten the Discussion (Section 7) to synthesize the main insights. Section 7.1 now identifies four recurring failure modes across method categories. Section 7.2 connects these failure modes to method-level assumptions through the new Table 5. Section 7.3 discusses the broader methodological implications for the field, including how motivating observations are validated and how mitigation success is measured. The Conclusion (Section 8) has also been rewritten around four explicit takeaways and four open challenges.
>
> ### Comments to Strengthen
>
> **Comment 1.4:** *Better connect method assumptions to observed failure cases.*
>
> **Response:** We have added this connection explicitly in Section 7.2 ("From Method Assumptions to Empirical Shortcomings"). The new Table 5 maps each evaluated method's key assumption(s) to the most directly relevant empirical shortcoming, organized by the method categories introduced in Sections 3 and 4. The surrounding prose draws cross-cutting patterns that are not visible from any single method description. We also avoid overclaiming mechanistic causation: the mapping is framed as interpretive because proving causation would require method-specific ablations beyond the scope of this benchmark.
>
> **Comment 1.5:** *Improve consistency in the level of detail across method descriptions.*
>
> **Response:** We have revised Sections 3 and 4 to make the method descriptions more consistent. Each description now follows the same general structure: (i) the motivating observation; (ii) the optimization problem, with the relevant equation displayed when appropriate; (iii) key assumptions; and (iv) limitations. We also corrected naming and category inconsistencies carried over from the previous draft.
>
> **Comment 1.6:** *Structure the discussion of limitations more clearly.*
>
> **Response:** We now organize limitations in two complementary ways. Per-method limitations remain within Sections 3 and 4, where they can be evaluated in the context of each method. Cross-cutting limitations are synthesized in Section 7 along three axes: recurring failure mode (Section 7.1), underlying design-time assumption (Section 7.2 and Table 5), and broader methodological implication for the field (Section 7.3). This separation avoids the previous structure, in which per-method and cross-cutting limitations were interleaved.

---

> > ### Comment · Reviewer_wNp6 · 2026-05-15
> >
> > Thank you for the detailed response and revisions. The updated manuscript addresses my main concerns, particularly regarding the paper’s positioning, synthesis of insights, and connection between assumptions and empirical findings.

---

### Decision · Action_Editor_bo4Q · 2026-05-25

**Recommendation:** Accept as is

**Audience:**

Yes

**Audience Explanation:**

Backdoor Attack is an important topic. A survey will be helpful for readers to know the current status of this field.

**Claims And Evidence:**

Yes

**Claims Explanation:**

All reviewers agree to accept the paper.